# SkipGPT: Each Token is One of a Kind

**Anhao Zhao** [1,2]  **Fanghua Ye** [3]  **Yingqi Fan** [1]  **Junlong Tong** [1,4]  **Jing Xiong** [5]  **Zhiwei Fei** [6]  **Hui Su** [7]  **Xiaoyu Shen** [1]

## Abstract

Large language models (LLMs) achieve remarkable performance across tasks but incur substantial computational costs due to their deep, multilayered architectures. Layer pruning has emerged as a strategy to alleviate these inefficiencies, but conventional static pruning methods overlook two critical dynamics inherent to LLM inference: (1) *horizontal dynamics*, where token-level heterogeneity demands context-aware pruning decisions, and (2) *vertical dynamics*, where the distinct functional roles of MLP and self-attention layers necessitate component-specific pruning policies. We introduce **SkipGPT**, a dynamic layer pruning framework designed to optimize computational resource allocation through two core innovations: (1) global token-aware routing to prioritize critical tokens and (2) decoupled pruning policies for MLP and self-attention components. To mitigate training instability, we propose a two-stage optimization paradigm: first, a disentangled training phase that learns routing strategies via soft parameterization to avoid premature pruning decisions, followed by parameter-efficient LoRA fine-tuning to restore performance impacted by layer removal. Extensive experiments demonstrate that SkipGPT reduces over 40% model parameters while matching or exceeding the performance of the original dense model across benchmarks. By harmonizing dynamic efficiency with preserved expressivity, SkipGPT advances the practical deployment of scalable, resource-aware LLMs. Our code is publicly available at: https://github.com/EIT-NLP/SkipGPT.

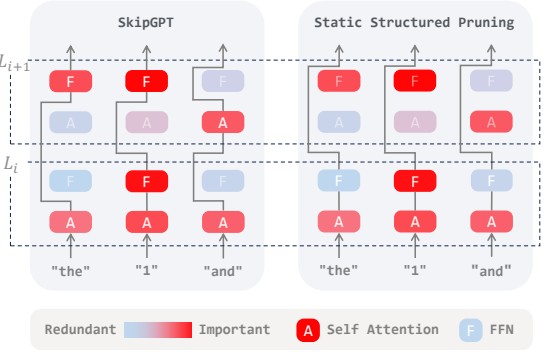

*Figure 1.* An overview of SkipGPT. Unlike conventional static structured pruning, SkipGPT dynamically prunes layers by considering both horizontal and vertical dynamics. In **horizontal dynamics**, different tokens receive varying computational allocations. In **vertical dynamics**, the MLP and attention modules are decoupled to account for their distinct roles within each layer.

## 1. Introduction

Large language models (LLMs) are built on a layer-wise Transformer architecture, where each layer consists of a self-attention mechanism followed by a multi-layer perceptron (MLP) (Vaswani et al., 2017). Scaling up model size has driven significant breakthroughs across a wide range of tasks (Brown, 2020; Bommasani et al., 2022; Wei et al., 2023; Zhao et al., 2024; Xin et al., 2025; Chen et al., 2025). However, this progress comes at a steep computational cost, requiring vast resources for inference (Chowdhery et al., 2022; Wan et al., 2024; OpenAI et al., 2024). In contrast, the human brain—despite its 100 trillion synaptic connections, far surpassing even the largest LLMs—operates efficiently on just 30 watts of power (Bartol et al., 2015; Samsi et al., 2023). This stark disparity underscores a fundamental inefficiency in current LLM architectures, highlighting the gap between artificial intelligence and human cognition.

Given their sequential layer-wise structure, LLMs struggle to fully leverage parallelism, even with abundant computational resources. This limitation makes layer pruning a crucial strategy for accelerating inference and improving efficiency (Men et al., 2024; Kim et al., 2024; Gromov et al., 2024; Chen et al., 2024b). While existing pruning methods offer some improvements, they often overlook two key aspects of pruning dynamics (see Figure 1):

---

[1]Ningbo Key Laboratory of Spatial Intelligence and Digital Derivative, Institute of Digital Twin, Eastern Institute of Technology, Ningbo [2]Southwest Jiaotong University [3]Tencent Inc. [4]Shanghai Jiao Tong University [5]The University of Hong Kong [6]Nanjing University [7]Meituan Inc.. Correspondence to: Fanghua Ye <fanghua.ye.21@gmail.com>, Xiaoyu Shen <xyshen@eitech.edu.cn>.

*Proceedings of the 42nd International Conference on Machine Learning*, Vancouver, Canada. PMLR 267, 2025. Copyright 2025 by the author(s).

1. **Horizontal Dynamics**: Different tokens in an input sequence require varying levels of computation. Current methods either allocate resources to the top-$k$ most relevant tokens per layer (Raposo et al., 2024; Zeng et al., 2023) or enforce a fixed computation ratio across all tokens (Jiang et al., 2024). These rigid approaches fail to adapt to token complexity, leading to suboptimal efficiency. To address this, we introduce a global sparsity mechanism, allowing computation budgets to be flexibly distributed across the entire forward pass rather than imposing fixed layer-wise or token-wise constraints.

2. **Vertical Dynamics**: The MLP and self-attention components within each layer serve distinct functions, yet most pruning methods treat them uniformly. Research suggests that MLPs function like localized neural processes, capturing task-specific interactions (Geva et al., 2021; Meng et al., 2023; Merullo et al., 2024), whereas attention mechanisms resemble higher-level cognitive functions, managing contextual relevance and information flow (Olsson et al., 2022; Kobayashi et al., 2020). Inspired by how the human brain activates different regions for different tasks, we propose decoupling the pruning of MLP and self-attention, enabling more targeted and efficient computation reduction.

To achieve dynamic pruning, a few recent works have explored adaptive computation methods that introduce a router at each Transformer layer (Zeng et al., 2023; Raposo et al., 2024; Jiang et al., 2024). This router functions as a decision module, determining whether specific network units should be executed or skipped during inference. However, these approaches typically optimize the router and model parameters simultaneously in a joint training paradigm, similar to Mixture-of-Experts (MoE) (Lepikhin et al., 2020; Cai et al., 2024; Zhu et al., 2024). However, this method fails to account for a fundamental difference between pruning and pretraining—*in the pruning context, the router starts from random initialization, while the model parameters have already converged to an optimal or locally optimal distribution through extensive pretraining*. This mismatch may make joint training unstable and prevent dynamic pruning from reaching its full potential.

In this work, we first provide empirical evidence demonstrating the significance of both horizontal dynamics and vertical dynamics in LLMs. To incorporate these two dynamics, we propose **SkipGPT**, a novel approach that dynamically prunes layers on a per-token basis, *adapting the pruning process to the complexity of each token*. Furthermore, SkipGPT *decouples the MLP and self-attention components within each layer*, enabling more granular control over which parts of the model are pruned, thereby optimizing both computational efficiency and model performance. To fully unlock the potential of dynamic pruning, we introduce a **Two-stage**

**Training Paradigm**. First, in **Router Tuning**, we freeze the model parameters and optimize only the router, allowing it to identify the most critical computations without disrupting the model's pretrained knowledge. Second, in **LoRA Fine-Tuning**, we freeze the router and fine-tune the model using LoRA to compensate for any performance degradation caused by pruning. Our results establish SkipGPT as a highly effective pruning strategy—enabling the pruned model to **fully restore its performance**, even **surpassing** the original model, despite a 40% reduction in parameters.

Furthermore, since router tuning does not modify the pretrained model parameters, it allows for a direct analysis of module importance in the original model. Through detailed router behavior analyses, we uncover two key insights: *(1) Attention modules exhibit greater redundancy than MLP modules. (2) As context length increases, later tokens demand more attention computation but less MLP processing.* These findings highlight inherent inefficiencies in current large model architectures, providing valuable insights for future architectural design and inference optimization.

## 2. Motivation

**Measuring module importance using cosine similarity** Recent work has demonstrated that the cosine similarity between a module's input and output serves as a reliable metric for evaluating the importance of each module (Men et al., 2024; Gromov et al., 2024). Notably, these studies often define a "module" as an entire Transformer layer. *To enable more fine-grained analysis, we refine the definition of a module to refer to either an MLP block or an attention block within a layer.* The underlying hypothesis of using cosine similarity is that redundant modules generate outputs that closely resemble their inputs, indicating minimal transformation. Conversely, important modules substantially alter their inputs, suggesting they play a critical role in the model and should be preserved. Prior approaches typically average the cosine similarity over all tokens to derive a general importance metric for each module. However, this aggregation may obscure the variability of module importance across different tokens and contexts. To better understand this variability, we analyze the cosine similarity of each module at the token level. Specifically, the cosine similarity $\mathbf{C}_{i,t}$ of the $i^{th}$ module at token $t$ is computed as:

$$\mathbf{C}_{i,t} = \frac{x_{i,t}^T x_{i+1,t}}{\|x_{i,t}\|_2 \|x_{i+1,t}\|_2}, \tag{1}$$

where $\| \cdot \|_2$ is the L2-norm and $x_{i,t}$ denotes the hidden state before module $i$ at token $t$. To illustrate how module importance varies, we conduct a case study using a randomly selected sentence from the BookCorpus dataset (Zhu et al., 2015). We analyze cosine similarity distributions across 15 consecutive tokens in LLaMA-2-7B (Touvron et al., 2023a), with the results visualized in Figure 2.

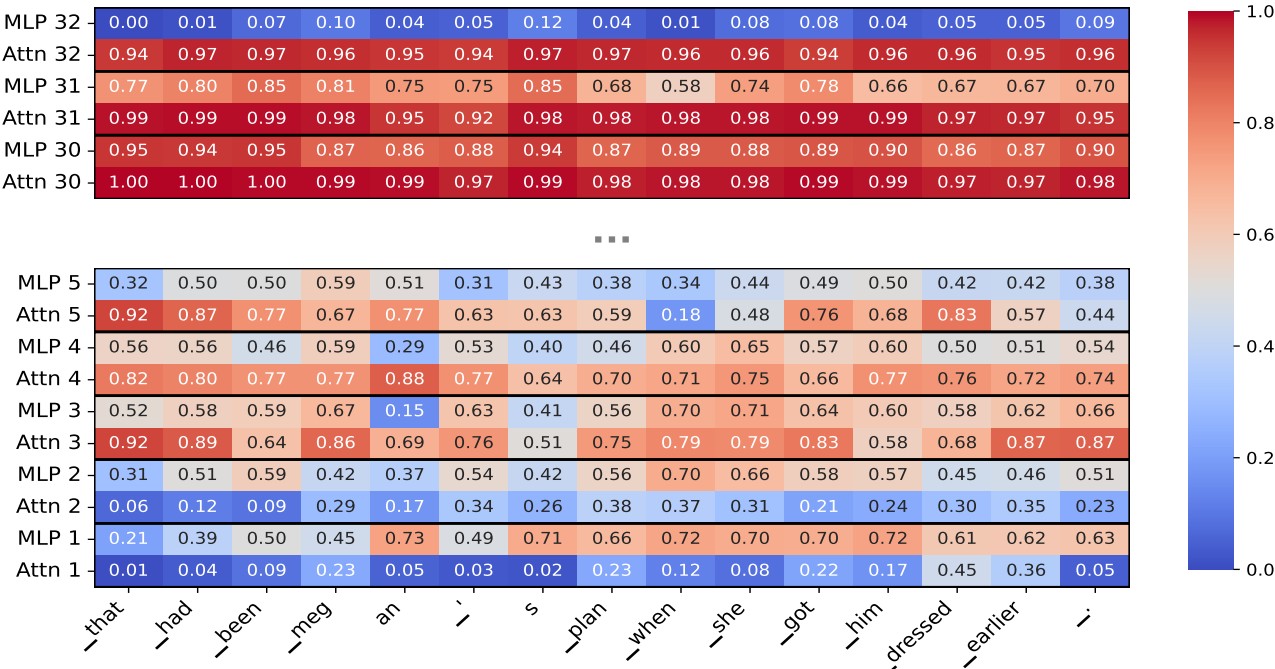

*Figure 2.* **Token-Wise Cosine Similarities Across Modules in LLaMA-2-7B,** which consists of 32 layers, corresponding to 64 modules in total. Due to space constraints, we showcase only the results for the initial and final modules. Higher values indicate greater redundancy.

**The Necessity of Vertical Dynamics** Existing pruning methods, whether dynamic or static, typically treat an entire transformer layer as the smallest pruning unit. However, as illustrated in Figure 2, we find that within the same layer, the cosine similarity distributions of attention and MLP modules can vary significantly. For example, in the second layer, the cosine similarity of most tokens in the attention module ranges from 0.1 to 0.4, while the MLP module predominantly falls within the range of 0.4 to 0.7. This indicates that for this layer, the MLP module is almost universally more redundant than the attention module. In the final layer, however, the situation is entirely reversed: all attention modules exhibit cosine similarity values between 0.9 and 1, while MLP modules fall within a completely different range, from 0 to 0.2, making the MLP module far more critical than the attention module at this stage. These findings reveal that *even within the same layer, redundancy levels between attention and MLP can vary and shift across layers*, underscoring the necessity of decoupling attention and MLP modules for more effective pruning.

**The Necessity of Horizontal Dynamics** Static pruning methods rely on two key assumptions: (1) the distribution of important modules is uniform across all tokens, and (2) each token is associated with the same number of important modules (Men et al., 2024; Kim et al., 2024; Gromov et al., 2024). Existing dynamic pruning methods also adopt these assumptions when setting compute budgets (Zeng et al., 2023; Raposo et al., 2024; Jiang et al., 2024). However,

these assumptions do not hold in practice. First, *module importance varies across layers*. For instance, at a cosine similarity threshold of 0.8, the attention module in layer 31 is redundant for 15 tokens, whereas in layer 4, it is redundant for only 3—clearly disproving uniform importance. Second, *token importance is not uniform within a sequence*. In our case study, we analyze the number of modules with a cosine similarity below 0.6—considering them significant—for each token. The results reveal that the token "plan" is associated with 13 important modules, while "an" has only 8, confirming that *different tokens require varying levels of computation*.

## 3. SkipGPT: Dynamic Layer Pruning

After establishing the necessity of horizontal and vertical dynamics, we now introduce *SkipGPT*, our proposed framework for dynamic layer pruning. In this section, we first outline the necessary preliminaries for understanding SkipGPT's optimization process, followed by an explanation of its sparsity mechanism, routing implementation, loss function, and finally, our two-stage training paradigm for stable and effective learning.

### 3.1. Preliminaries: Gumbel-Softmax and STE

**Gumbel-Softmax Reparametrization** To optimize SkipGPT's dynamic pruning decisions, we formulate the pruning process as a discrete optimization problem,

where each module (MLP or attention) is either executed or skipped based on its computed importance. However, directly optimizing discrete decisions is non-differentiable, making standard gradient-based optimization infeasible. The Gumbel-Softmax distribution is a continuous approximation of the categorical distribution, enabling differentiable sampling (Jang et al., 2022). This is achieved via **reparameterization**, which transforms discrete samples into differentiable continuous ones for gradient-based optimization. Let $\pi_1, \pi_2, \ldots, \pi_k$ represent the class probabilities of a $k$-class categorical distribution. To sample from this distribution, the *Gumbel-Max trick* (Gumbel, 1954; Maddison et al., 2015) selects the category with the highest value of $\log \pi_i + g_i$, where $g_i$ are i.i.d. samples from the Gumbel distribution $\text{Gumbel}(0, 1)$[1]:

$$z = \text{one\_hot}\left(\arg\max_i [g_i + \log \pi_i]\right). \quad (2)$$

Since $\arg\max$ is non-differentiable, the Gumbel-Softmax reparametrization replaces it with a softmax function, producing continuous samples approximating the categorical distribution:

$$y_i = \frac{\exp\left(\frac{\log \pi_i + g_i}{\tau}\right)}{\sum_{j=1}^{k} \exp\left(\frac{\log \pi_j + g_j}{\tau}\right)}, \quad i = 1, \ldots, k, \quad (3)$$

where $\tau$ controls the sharpness of the distribution. As $\tau \to 0$, the samples resemble one-hot vectors, recovering the original $\arg\max$ operation. This differentiable approximation enables standard backpropagation for optimization.

**Straight-Through Estimator** The Straight-Through (ST) Estimator (Bengio et al., 2013) enables discrete sampling while preserving differentiability for backpropagation. In the forward pass, we use Gumbel-Softmax to generate continuous samples. To discretize them, we apply $\arg\max$, but in the backward pass, gradients are computed as if using the continuous approximation. This is achieved via:

$$y_{\text{hard}} = y_{\text{hard}} - y_{\text{soft}} \cdot \text{detach()} + y_{\text{soft}}, \quad (4)$$

where $y_{\text{soft}}$ is the continuous Gumbel-Softmax sample. This ensures a **one-hot output while gradients follow $y_{\text{soft}}$**, enabling efficient optimization despite discrete sampling.

### 3.2. The Concept of Sparsity

To control the total FLOPs, we introduce the concept of sparsity, defined as *the fraction of module computations (attention or MLP) skipped in a forward pass, relative to the total computations in a fully dense transformer, accounting for all layers and sequence positions.* In our method,

sparsity is achieved through dynamic routing, where only a subset of modules is selected for computation in each forward pass. Specifically, assuming that in a forward pass, an $L$-layer LLM (which consists of $2L$ modules, each layer containing one attention module and one MLP module) processes a sequence of length $S$, a dense transformer would compute $2L \times S$ modules, corresponding to a sparsity of 0. When sparsity is greater than 0, only $(1 - \text{sparsity}) \times 2L \times S$ modules are computed.

Unlike previous work, which defines the compute budget by restricting computation to the top-$k$ tokens at each layer (Raposo et al., 2024; Zeng et al., 2023) or enforcing the same sparsity for each token (Jiang et al., 2024), we allow the computational load to be dynamically allocated across both width (the number of tokens participating in computation at each layer) and height (the number of modules involved in the computation for each token).

### 3.3. Routing Implementation

To enable dynamic allocation, SkipGPT assigns a router before each individual module. Specifically, we route tokens to two computational paths: *(1) self-attention (for the attention router) or FFN modules (for the FFN router), and (2) a residual connection.* The latter is computationally inexpensive, producing an output determined entirely by the input, while the former incurs a high computational cost.

Suppose that we have a token embedding $x_l$ prior to the $l$-th transformer module $f_l$, where this module can either be a self-attention or an FFN. Before passing through the module $f_l$, $x_l$ first undergoes a router function, which is a simple linear projection, yielding a categorical distribution $r_l = \mathbf{W}_\theta^T x_l \in \mathbb{R}^2$, where the first element represents the probability of skipping the module, and the second element represents the probability of executing it.

Once this categorical distribution is obtained, a natural routing strategy is the Top-1 routing. Specifically, we have:

$$x_{l+1} = \begin{cases} r_l[1] \cdot (f_l(x_l) + x_l), & \text{if } \arg\max r_l = 1, \\ r_l[0] \cdot x_l, & \text{otherwise,} \end{cases} \quad (5)$$

such that the gradients can be backpropagated. *However, this routing strategy challenges precise sparsity control, as $r_l$ is merely a soft approximation of binary selection.*

By leveraging Gumbel-Softmax and the ST Gumbel Estimator, we effectively address this issue. Specifically, after applying Gumbel-Softmax and the ST Gumbel Estimator to $r_l$, we obtain a one-hot vector $g_l \in \{0, 1\}^2$. Thus, the input to the next module is computed as:

$$x_{l+1} = g_l[1] \cdot (f_l(x_l) + x_l) + g_l[0] \cdot x_l. \quad (6)$$

During the forward pass, **discrete binary values** are sampled, ensuring clear pruning decisions. In the backward pass,

---

[1] $g$ can be obtained via inverse transform sampling: $u \sim \text{Uniform}(0, 1)$, $g = -\log(-\log(u))$.

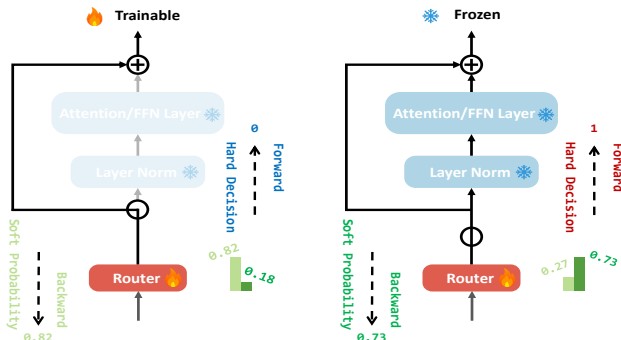

*Figure 3.* Illustration of the forward and backward passes during the router tuning stage. In the forward pass, the router makes hard decisions to either execute (1) or skip (0). In the backward pass, the gradients are propagated back using soft probabilities.

soft probabilities facilitate gradient propagation, allowing the router weights to be updated effectively.

With this routing design, in principle, we can train the router independently without altering the model parameters to obtain the optimal routing solution for a given sparsity ratio.

### 3.4. Loss Function

With our definition in Section 3.2, the sparsity $r$ is given by:

$$r = \frac{\sum_{t,l} g_l^t[0]}{S \times 2L}, \qquad (7)$$

where $t$ is the token index, $l$ represents the module index, $L$ is the total number of layers in the LLM, and $S$ denotes the length of the sequence. To meet different computational demands, we introduce a sparsity regularization term:

$$\mathcal{L}_{\text{sparsity}} = |\mathcal{T} - r|, \qquad (8)$$

where $|\cdot|$ denotes absolute value, and $\mathcal{T}$ is the user-defined target sparsity. The overall loss function is then given by:

$$\mathcal{L}_{\text{all}} = \mathcal{L}_{\text{lm}} + \alpha \mathcal{L}_{\text{sparsity}}, \qquad (9)$$

where $\mathcal{L}_{\text{lm}}$ is the standard language modeling loss, which represents the negative log-likelihood of predicting the next token on average. The hyperparameter $\alpha$ controls the strength of the sparsity penalty in the overall loss function.[2]

### 3.5. Two-stage Training Paradigm

As the router starts from random initialization and the LLM has already been pretrained, direct training can lead to unstable and suboptimal pruning decisions. To address this issue, we propose a two-stage training paradigm.

---

[2]If $\alpha$ is too small, it may fail to enforce the desired sparsity. If $\alpha$ is too large, the model may compromise the optimization of $\mathcal{L}_{\text{lm}}$. Based on our experiments, a value of 8 works well.

**Router Tuning** In the initial stage of training, we focus exclusively on tuning the router while keeping all other model parameters frozen. This stage is highly efficient, as it requires adding only a lightweight linear layer before each module, with all router parameters combined accounting for just 0.007% of the total parameters in LLaMA2-7B. Through this stage, we achieve **over 90%** of the model's original performance even after **discarding 25%** of the parameters, as shown in Table 1. The forward and backward passes during router tuning are illustrated in Figure 3.

**LoRA Fine-Tuning** *(Optional)* While the router tuning stage is sufficient to preserve most of the model's performance, we provide an optional LoRA fine-tuning stage for those aiming to fully restore performance to the **original model level**. Low-Rank Adaptation (LoRA) (Hu et al., 2021) enables efficient refinement of LLMs with minimal computational overhead. Previous works, such as Ma et al. (2023); Kim et al. (2024), have demonstrated LoRA's effectiveness in enhancing statically pruned models. In this study, we show that LoRA can also effectively recover the performance of dynamically pruned models. For a clearer understanding of the entire training process, please refer to the algorithm illustrated in Appendix B.

## 4. Experimental Configuration

### 4.1. Models and Benchmarks

**Models** We conduct experiments utilizing *LLaMA2-7B*, *LLaMA2-13B*, and *LLaMA3.1-8B* (Touvron et al., 2023a;b; Dubey et al., 2024).

**Data** During both router and LoRA tuning, we use the *RedPajama-Data-1T-Sample* dataset (Computer, 2023)[3], which contains 850,000 samples (1 billion tokens) truncated to 4096 tokens each. This dataset serves two roles: (1) as a calibration set (100 random samples) to compute block-level significance for pruning redundant layers (static methods), and (2) as a training set for dynamic methods and for recovering static method performance (the specific details of static and dynamic methods will be introduced later in Section 4.2).

**Training Procedure** Each model is trained for 10,000 steps with next-token prediction, using a batch size of 16 in both router and LoRA tuning. In the router tuning stage, we use a constant learning rate of 2e-3[4]. Additionally, the softmax temperature $\tau$ of the Gumbel-Softmax is linearly annealed from 5 to 1. In the LoRA tuning stage, the learning

---

[3]https://huggingface.co/datasets/togethercomputer/RedPajama-Data-1T-Sample

[4]A grid-search was conducted to determine that a learning rate of 2e-3 optimally ensures training stability.

| Model | Method | Ratio | OBQA AccNorm | WinoGrande Acc | PIQA Acc | HeSw AccNorm | BoolQ Acc | ARC-E AccNorm | ARC-C AccNorm | Avg. Acc.↑ | WT2 PPL | PTB PPL | Avg. PPL↓ |
|---|---|---|---|---|---|---|---|---|---|---|---|---|---|
| **LLaMA2-7B** | Dense | 0.00% | 44.20 | 74.19 | 78.07 | 78.93 | 71.62 | 81.36 | 52.47 | 68.69 | 5.47 | 20.83 | 13.15 |
| | ShortGPT | 25.0% | 34.80 | 68.43 | 67.68 | 60.77 | 62.17 | 59.34 | 38.40 | 55.94 | 25.42 | 70.97 | 48.20 |
| | Shortened-PPL | 25.0% | 33.60 | 52.88 | 70.40 | 55.12 | 61.07 | 55.05 | 29.44 | 51.08 | 11.15 | 49.07 | 30.11 |
| | Shortened-Taylor | 25.0% | 35.40 | 66.30 | 66.97 | 59.90 | 62.17 | 59.76 | 38.65 | 55.59 | 23.75 | 69.60 | 46.68 |
| | Joint Layer Drop | 23.9% | 38.60 | 72.38 | 70.08 | 67.93 | 40.37 | 65.57 | 44.54 | 57.07 | 29.19 | 64.69 | 46.94 |
| | LaCo | 25.0% | 36.60 | 67.32 | 65.72 | 62.76 | 74.37 | 58.92 | 38.14 | 57.69 | 18.96 | 47.58 | 33.27 |
| | LLM-Pruner | 25.3% | 39.00 | 58.56 | 73.45 | 60.77 | 54.71 | 58.63 | 37.03 | 54.59 | 14.10 | 65.09 | 39.60 |
| | SliceGPT | 25.4% | 35.40 | 65.82 | 66.38 | 53.43 | 50.43 | 69.57 | 40.10 | 54.45 | 7.56 | 76.29 | 41.93 |
| | SkipGPT-RT | 25.5% | 39.60 | 63.54 | 72.20 | 70.96 | 68.81 | 76.52 | 44.37 | 62.29 | 7.81 | 30.58 | 19.20 |
| **LLaMA2-13B** | Dense | 0.00% | 45.20 | 76.16 | 79.11 | 82.23 | 80.52 | 84.68 | 59.47 | 72.48 | 4.88 | 28.92 | 16.90 |
| | ShortGPT | 25.0% | 40.60 | 70.80 | 71.38 | 72.59 | 62.69 | 69.19 | 45.31 | 61.79 | 20.05 | 49.52 | 34.79 |
| | Shortened-PPL | 25.0% | 39.40 | 67.17 | 73.12 | 69.31 | 62.57 | 69.19 | 41.13 | 60.27 | 8.45 | 54.62 | 31.54 |
| | Shortened-Taylor | 25.0% | 41.80 | 70.80 | 70.78 | 62.58 | 38.10 | 61.99 | 38.31 | 54.90 | 24.46 | 59.34 | 41.90 |
| | Joint Layer Drop | 24.3% | 41.80 | 72.66 | 73.37 | 74.43 | 70.24 | 71.25 | 46.50 | 64.69 | 13.31 | 42.63 | 27.97 |
| | LaCo | 25.0% | 38.80 | 62.75 | 72.74 | 63.11 | 44.46 | 62.84 | 35.07 | 54.25 | 13.40 | 53.92 | 33.66 |
| | LLM-Pruner | 24.9% | 44.00 | 65.82 | 76.99 | 74.15 | 59.88 | 72.60 | 45.82 | 62.75 | 9.82 | 71.49 | 40.66 |
| | SliceGPT | 24.7% | 38.60 | 68.43 | 64.42 | 50.97 | 38.87 | 69.36 | 40.36 | 53.00 | 7.43 | 99.89 | 53.66 |
| | SkipGPT-RT | 25.4% | 46.00 | 71.90 | 76.88 | 74.33 | 74.37 | 77.69 | 47.08 | 66.89 | 6.78 | 41.69 | 24.24 |
| **LLaMA3.1-8B** | Dense | 0.00% | 44.80 | 77.51 | 80.03 | 81.95 | 82.14 | 84.85 | 57.59 | 72.69 | 6.24 | 10.58 | 8.41 |
| | ShortGPT | 25.0% | 28.00 | 54.14 | 58.76 | 31.50 | 37.77 | 38.05 | 31.40 | 39.95 | 2796.24 | 2799.46 | 2797.85 |
| | Shortened-PPL | 25.0% | 33.60 | 53.51 | 71.87 | 57.98 | 42.08 | 57.07 | 31.74 | 49.69 | 15.00 | 23.86 | 19.43 |
| | Shortened-Taylor | 25.0% | 28.20 | 54.06 | 58.87 | 31.53 | 37.77 | 38.05 | 31.31 | 39.97 | 2690.34 | 2793.25 | 2,741.80 |
| | Joint Layer Drop | 24.2% | 33.00 | 55.56 | 61.26 | 42.26 | 37.31 | 35.06 | 30.12 | 42.08 | 32.32 | 51.40 | 41.86 |
| | LaCo | 24.5% | 31.20 | 65.11 | 66.16 | 55.77 | 71.13 | 37.03 | | 53.58 | 30.14 | 50.35 | 40.25 |
| | LLM-Pruner | 24.5% | 37.20 | 56.99 | 72.03 | 54.75 | 56.79 | 51.22 | 31.31 | 51.47 | 25.21 | 45.59 | 35.40 |
| | SliceGPT | 24.6% | 30.40 | 55.17 | 57.83 | 38.19 | 37.83 | 38.64 | 25.85 | 40.56 | 17.25 | 63.13 | 40.19 |
| | SkipGPT-RT | 25.5% | 44.20 | 75.69 | 78.07 | 76.87 | 74.06 | 82.44 | 53.41 | 69.25 | 16.47 | 26.91 | 21.69 |
| **LLaMA3.1-8B** | Dense | 0.00% | 44.80 | 77.51 | 80.03 | 81.95 | 82.14 | 84.85 | 57.59 | 72.69 | 6.24 | 10.58 | 8.41 |
| | ShortGPT | 40.6% | 30.00 | 57.62 | 58.54 | 34.55 | 62.29 | 34.81 | 29.10 | 43.84 | 79856.66 | 125507.27 | 102681.97 |
| | Shortened-PPL | 40.6% | 27.00 | 52.41 | 61.48 | 41.74 | 57.13 | 39.69 | 26.45 | 43.70 | 157.01 | 196.04 | 176.53 |
| | Shortened-Taylor | 40.6% | 28.80 | 53.20 | 59.68 | 37.89 | 58.53 | 35.19 | 29.69 | 43.28 | 78424.35 | 125435.43 | 101,929.90 |
| | Joint Layer Drop | 39.9% | 27.60 | 50.12 | 53.48 | 26.65 | 37.86 | 26.35 | 25.77 | 35.40 | 251.52 | 341.25 | 296.39 |
| | LaCo | 40.7% | 27.60 | 51.54 | 56.31 | 31.53 | 55.11 | 30.22 | 25.85 | 39.74 | 269.24 | 392.50 | 330.87 |
| | LLM-Pruner | 39.9% | 29.20 | 51.30 | 64.09 | 36.23 | 50.52 | 36.87 | 23.46 | 41.67 | 152.98 | 161.49 | 157.24 |
| | SliceGPT | 39.9% | 25.60 | 51.62 | 53.92 | 30.64 | 37.83 | 31.82 | 22.10 | 36.22 | 143.24 | 183.21 | 163.23 |
| | SkipGPT-RT | 40.2% | 38.00 | 59.35 | 73.34 | 64.36 | 60.37 | 77.53 | 45.65 | 59.80 | 71.25 | 48.05 | 59.65 |

*Table 1.* Comparison of SkipGPT-RT and static pruning baselines without LoRA, where the ratio denotes the proportion of parameters (averaged per token) that do not participate in computations relative to the original model's total parameter count.

rate is set to 2e-4 with a warmup ratio of 0.1 and a cosine learning rate scheduler. The AdamW optimizer (Loshchilov & Hutter, 2019) with $\beta_1 = 0.9$ and $\beta_2 = 0.95$ is used for gradient backpropagation.

**Benchmarks**  Following Touvron et al. (2023a), we evaluate accuracy scores on a variety of commonsense reasoning datasets, including *BoolQ* (Clark et al., 2019), *PIQA* (Bisk et al., 2020), *HellaSwag (HeSw)* (Zellers et al., 2019), *WinoGrande* (Sakaguchi et al., 2021), *ARC-easy (ARC-E)* (Clark et al., 2018), *ARC-challenge (ARC-C)* (Clark et al., 2018), and *OpenbookQA (OBQA)* (Mihaylov et al., 2018), using the *lm-evaluation-harness* (Gao et al., 2024). Additionally, we report zero-shot PPL scores on *WikiText2 (WT2)* (Merity et al., 2016) and *PTB* (Marcus et al., 1993).

### 4.2. Baseline

**Static Pruning Baselines**  Static pruning involves the permanent removal of redundant model components. *ShortGPT* (Men et al., 2024) removes redundant layers based on Block Influence (BI) scores. *Shortened LLaMA* (Kim et al., 2024) prunes layers using PPL and Taylor Expansion, resulting in two variants: *Shortened-PPL* and *Shortened-Taylor*.

*LaCo* (Yang et al., 2024) collapses layers progressively from deep to shallow, employing a threshold to prevent excessive merging. *Joint Layer Drop* (He et al., 2024) is a fine-grained variant of ShortGPT that separately removes attention and MLP layers based on the same BI metric. *LLM-Pruner* (Ma et al., 2023) selectively removes non-critical structures using gradient-based criteria. *SliceGPT* (Ashkboos et al., 2024) shrinks the embedding dimension by replacing large weight matrices with smaller dense matrices. Unlike static pruning methods, SkipGPT considers horizontal dynamics, offering a more adaptive approach.

**Dynamic Pruning Baselines**  Dynamic pruning allows for the selective activation of model components during inference. MoD (Raposo et al., 2024) employs a top-$k$ routing mechanism to dynamically activate layers for each token. To ensure a fair comparison, we introduce a new variant, *MoD-D*, which differs from MoD by decoupling attention and MLP modules, whereas MoD treats them as a single unit. *D-LLM* (Jiang et al., 2024) introduces a dynamic decision module at each transformer layer, determining whether a layer should be executed or skipped. Additionally, we include *SkipGPT-Joint*, which adopts a joint training paradigm instead of two-stage training. From the perspective of hori-

zontal dynamics, MoD-D allocates its computation budget layer-wise, while D-LLM allocates it token-wise. Regarding vertical dynamics, D-LLM does not decouple attention and MLP modules. Please refer to Appendix C for detailed descriptions of these baselines.

## 5. Results

This section provides a detailed analysis of the models after each training stage. For clarity, we define the model obtained after the first stage (Router Tuning) as **SkipGPT-RT** and the final model after the second stage (LoRA Fine-Tuning) as **SkipGPT-RT-L**.

### 5.1. Comparison Between SkipGPT-RT and Baselines

We present the accuracy and PPL for baseline pruning methods and SkipGPT-RT in Table 1. For all models we evaluate, including LLaMA2-7B, LLaMA2-13B, and LLaMA3.1-8B, the attention module contains approximately half as many parameters as the MLP module. *Thus, in Joint Layer Drop and SkipGPT-RT, we prune proportionally more modules to maintain a consistent average parameter ratio across methods.* For long-context tasks, the FLOPs of the attention module surpass those of the MLP, making SkipGPT-RT achieve lower computational overhead compared to other baselines. For LLaMA2-7B and LLaMA2-13B, we report evaluation results under a 25% parameter reduction setting, while for LLaMA3.1-8B, we provide results for both 25% and 40% parameter reduction. Additional results under varying pruning ratios are presented in Appendix D.

The results demonstrate that with router tuning alone (without modifying model parameters), SkipGPT-RT significantly outperforms baseline methods. Specifically, for LLaMA2-7B and LLaMA2-13B, SkipGPT-RT retains over 90% of the dense model's performance even under 25% parameter pruning. For LLaMA3.1-8B, it retains over 95% performance at 25% pruning and over 80% performance at 40% pruning — a level at which nearly all baselines collapse catastrophically. Notably, while router tuning involves training the router parameters, unlike static pruning, which permanently removes layers based on predefined metrics, its cost is minimal. This is because the router parameters constitute less than 0.01% of the total model parameters and converge rapidly (Figure 4). *In practice, the tuning process requires only a single A800 (80GB) GPU and completes within four hours.* Furthermore, the results in Table 1 highlight three key observations:

- Layer-pruning methods—such as ShortGPT, LaCo, Shortened LLaMA, and Joint Layer Drop—generally match or surpass embedding dimension reduction approaches like LLM-Pruner and SliceGPT. *This suggests that LLMs have greater redundancy in depth than width*, aligning

with prior work (Men et al., 2024). This strongly supports focusing on depth pruning when designing SkipGPT.

- Joint Layer Drop, the fine-grained variant of ShortGPT that independently removes attention and MLP, outperforms ShortGPT in most cases. This result validates our motivation for emphasizing **the necessity of Vertical Dynamics** to achieve more effective pruning.

- SkipGPT-RT significantly outperforms Joint Layer Drop across all models in accuracy (reflecting LLMs' ability as general-purpose task solvers) and perplexity (involving the capability to generate coherent and fluent sentences). This strongly supports our motivation for highlighting **the necessity of horizontal dynamics**.

### 5.2. Comparison between SkipGPT-RT-L and Baselines

**SkipGPT-RT-L ≈ original model performance**  Figure 4 shows training curves for SkipGPT-RT, SkipGPT-RT-L, static pruning methods with LoRA, and joint router+LoRA training for dynamic pruning for LLaMA2-7B and LLaMA2-13B. Results are summarized in Table 2. Notably, SkipGPT-RT achieves performance comparable to the best fine-tuned baseline using router tuning alone. After LoRA fine-tuning, our method not only fully restores the model's performance to the original level but even surpasses it (without fine-tuning), ranking second only to directly applying LoRA to the original model for LLaMA2-7B. *For those aiming to quickly build a high-performing pruned model, router tuning is highly effective. Meanwhile, to fully match or exceed the original model, LoRA fine-tuning is an excellent choice.* It is worth reiterating that the entire process for all models requires only a single A800 GPU.

**The Effectiveness of Two-Stage Training Paradigm**  We designate *SkipGPT-Joint*, which refers to the variant where both the router parameters and model parameters are trained simultaneously, as the ablation baseline for the two-stage training paradigm. As shown in Table 2, its performance is significantly worse than SkipGPT. While it may seem intuitive that directly adopting MoE's joint training paradigm could outperform two-stage training—since the router and LoRA parameters can gradually adapt to each other's representations—the results tell a different story. Not only does SkipGPT-Joint underperform, but dynamic pruning methods following the joint training paradigm consistently fail to achieve satisfactory results. This finding reinforces our argument in Section 3.5: in the joint training paradigm, the randomly initialized router forces model parameters to adapt early on to a suboptimal, random routing strategy. This misalignment disrupts the model's established parameter distribution, making it increasingly difficult for the router to identify critical modules. Over time, this leads to a reinforcing feedback loop: as the model adapts to poor routing,

| Model | Method | Ratio | OBQA AccNorm | WinoGrande Acc | PIQA Acc | HeSw AccNorm | BoolQ Acc | ARC-E AccNorm | ARC-C AccNorm | Avg. Acc.↑ | WT2 PPL | PTB PPL | Avg. PPL↓ |
|---|---|---|---|---|---|---|---|---|---|---|---|---|---|
| **LLaMA2-7B** | Dense | 0.00% | 44.20 | 74.19 | 78.07 | 78.93 | 71.62 | 81.36 | 52.47 | 68.69 | 5.47 | 20.83 | 13.15 |
| | Dense, LoRA | 0.00% | 44.80 | 74.27 | 78.02 | 78.96 | 79.02 | 81.73 | 53.07 | 69.98 | 5.48 | 20.58 | 13.03 |
| | ShortGPT, LoRA | 25.0% | 39.40 | 71.74 | 71.38 | 70.12 | 73.91 | 70.37 | 41.72 | 62.66 | 8.45 | 35.96 | 22.21 |
| | Shortened-PPL, LoRA | 25.0% | 36.40 | 57.38 | 74.59 | 64.58 | 63.21 | 69.70 | 39.16 | 57.86 | 7.71 | 31.26 | 19.49 |
| | Shortened-Taylor, LoRA | 25.0% | 37.20 | 70.24 | 71.38 | 70.38 | 72.35 | 70.24 | 43.43 | 62.17 | 8.38 | 33.60 | 20.99 |
| | Joint Layer Drop, LoRA | 23.9% | 41.40 | 72.38 | 74.32 | 71.25 | 73.85 | 75.17 | 45.99 | 64.91 | 9.04 | 32.49 | 20.77 |
| | LaCo, LoRA | 25.0% | 39.20 | 70.40 | 70.95 | 69.40 | 76.73 | 69.74 | 42.66 | 62.73 | 8.38 | 28.77 | 18.58 |
| | LLM-Pruner, LoRA | 25.3% | 39.00 | 58.64 | 73.50 | 61.72 | 54.65 | 58.59 | 37.03 | 54.73 | 14.10 | 65.09 | 39.56 |
| | SliceGPT, LoRA | 25.4% | 38.60 | 67.88 | 72.58 | 69.34 | 71.13 | 75.29 | 45.39 | 62.89 | 7.19 | 38.91 | 23.05 |
| | MoD-D | 25.0% | 28.00 | 50.28 | 69.86 | 58.80 | 62.29 | 72.81 | 43.34 | 55.05 | 33.55 | 125.62 | 79.59 |
| | D-LLM | 25.6% | 25.60 | 52.88 | 51.14 | 40.81 | 57.98 | 38.09 | 27.13 | 41.95 | 34.75 | 143.85 | 89.30 |
| | SkipGPT-Joint | 25.3% | 24.40 | 50.83 | 50.98 | 62.81 | 43.36 | 62.83 | 38.23 | 48.16 | 10.12 | 346.05 | 178.09 |
| | SkipGPT-RT-L | 25.5% | **44.00** | **74.90** | **78.24** | **77.80** | **77.58** | **81.40** | **52.56** | **69.50** | **5.82** | **21.26** | **13.54** |
| **LLaMA2-13B** | Dense | 0.00% | 45.20 | 76.16 | 79.11 | 82.23 | 80.52 | 84.68 | 59.47 | 72.48 | 4.88 | 28.92 | 16.90 |
| | Dense, LoRA | 0.00% | 45.20 | 76.24 | 79.11 | 82.26 | 80.06 | 84.97 | 59.72 | 72.51 | 4.80 | 28.90 | 16.85 |
| | ShortGPT, LoRA | 25.0% | 42.20 | 74.90 | 74.54 | 76.60 | 64.37 | 76.94 | 49.74 | 65.61 | 6.78 | 34.48 | 20.63 |
| | Joint Layer Drop, LoRA | 24.3% | 44.80 | 73.56 | 77.48 | 77.20 | 77.83 | 79.97 | 51.96 | 68.97 | 5.58 | **27.34** | **16.46** |
| | LaCo, LoRA | 25.0% | 40.60 | 65.98 | 76.99 | 72.19 | 67.37 | 75.63 | 45.31 | 63.44 | 6.81 | 29.53 | 18.17 |
| | LLM-Pruner, LoRA | 24.9% | 44.00 | 65.82 | 76.99 | 74.13 | 59.88 | 72.64 | 45.82 | 62.76 | 9.83 | 71.49 | 40.66 |
| | MoD-D | 25.0% | 37.20 | 67.25 | 71.49 | 51.83 | 54.13 | 78.24 | 50.34 | 58.64 | 39.31 | 39.69 | 39.50 |
| | D-LLM | 25.2% | 28.00 | 57.06 | 52.18 | 60.35 | 62.78 | 58.54 | 35.92 | 50.69 | 13.09 | 31.69 | 22.39 |
| | SkipGPT-Joint | 25.2% | 38.00 | 62.90 | 60.83 | 71.44 | 68.32 | 66.50 | 46.93 | 59.27 | 8.49 | 27.97 | 18.23 |
| | SkipGPT-RT-L | 25.4% | **46.20** | **76.32** | **79.38** | **82.14** | **80.31** | **83.88** | **57.25** | **72.21** | **5.00** | 28.59 | 16.80 |
| **LLaMA3.1-8B** | Dense | 0.00% | 44.80 | 77.51 | 80.03 | 81.95 | 82.14 | 84.85 | 57.59 | 72.69 | 6.24 | 10.58 | 8.41 |
| | Dense, LoRA | 0.00% | 44.90 | 77.68 | 80.22 | 81.86 | 82.23 | 84.92 | 57.89 | 72.81 | 6.13 | 10.44 | 8.29 |
| | ShortGPT, LoRA | 25.0% | 38.40 | 70.96 | 73.94 | 69.23 | 72.05 | 68.01 | 43.86 | 62.35 | 11.13 | 16.64 | 13.89 |
| | Shortened-PPL, LoRA | 25.0% | 39.00 | 60.22 | 75.95 | 67.92 | 63.46 | 68.98 | 39.76 | 59.33 | 8.17 | 13.22 | 10.70 |
| | Shortened-Taylor, LoRA | 25.0% | 37.40 | 71.82 | 73.72 | 69.56 | 71.19 | 66.88 | 44.45 | 62.15 | 10.32 | 14.97 | 12.65 |
| | Joint Layer Drop, LoRA | 24.2% | 35.00 | 69.30 | 72.47 | 64.11 | 67.95 | 59.81 | 38.23 | 58.12 | 14.32 | 20.14 | 17.23 |
| | LaCo, LoRA | 24.5% | 36.40 | 69.46 | 71.93 | 66.50 | 76.24 | 64.73 | 41.13 | 60.91 | 10.77 | 17.23 | 14.00 |
| | LLM-Pruner, LoRA | 24.5% | 37.20 | 64.09 | 75.46 | 65.72 | 64.56 | 62.42 | 35.67 | 57.87 | 20.42 | 34.87 | 27.65 |
| | SliceGPT, LoRA | 24.6% | 34.40 | 61.56 | 66.70 | 56.96 | 72.39 | 50.08 | 31.48 | 53.37 | 9.22 | 26.42 | 17.82 |
| | MoD-D | 25.0% | 31.60 | 52.41 | 64.25 | 50.44 | 50.28 | 37.67 | 28.24 | 44.98 | 34.21 | 43.29 | 38.75 |
| | D-LLM | 25.0% | 30.20 | 52.49 | 57.40 | 37.64 | 50.36 | 37.12 | 28.16 | 41.91 | 40.12 | 132.44 | 86.28 |
| | SkipGPT-Joint | 25.3% | 31.50 | 52.34 | 60.13 | 50.87 | 50.74 | 37.21 | 28.66 | 44.49 | 9.87 | 31.28 | 20.58 |
| | SkipGPT-RT-L | 25.5% | **42.60** | **77.03** | **79.97** | **82.13** | **82.84** | **84.47** | **57.08** | **72.30** | **7.10** | **11.70** | **9.40** |
| **LLaMA3.1-8B** | Dense | 0.00% | 44.80 | 77.51 | 80.03 | 81.95 | 82.14 | 84.85 | 57.59 | 72.69 | 6.24 | 10.58 | 8.41 |
| | Dense, LoRA | 0.00% | 44.90 | 77.68 | 80.22 | 81.86 | 82.23 | 84.92 | 57.89 | 72.81 | 6.13 | 10.44 | 8.29 |
| | ShortGPT, LoRA | 40.6% | 32.00 | 67.32 | 68.61 | 58.43 | 65.38 | 53.37 | 35.32 | 54.35 | 18.35 | 30.65 | 24.50 |
| | Shortened-PPL, LoRA | 40.6% | 33.80 | 54.78 | 71.16 | 54.43 | 60.58 | 55.51 | 31.31 | 51.65 | 12.77 | 20.54 | 16.66 |
| | Shortened-Taylor, LoRA | 40.6% | 32.40 | 64.64 | 68.01 | 57.55 | 65.02 | 53.11 | 33.02 | 53.39 | 17.22 | 28.79 | 23.01 |
| | Joint Layer Drop, LoRA | 39.9% | 28.60 | 52.96 | 60.23 | 35.10 | 56.91 | 36.99 | 36.99 | 43.97 | 21.65 | 33.25 | 27.45 |
| | LaCo, LoRA | 40.7% | 30.20 | 56.27 | 65.13 | 43.99 | 62.05 | 43.27 | 26.79 | 46.81 | 16.43 | 24.66 | 20.55 |
| | LLM-Pruner, LoRA | 39.9% | 31.80 | 55.01 | 70.24 | 51.34 | 56.18 | 50.08 | 27.90 | 48.94 | 30.43 | 40.67 | 35.55 |
| | SliceGPT, LoRA | 39.9% | 28.20 | 55.41 | 60.61 | 44.15 | 67.52 | 40.70 | 25.34 | 45.99 | 14.87 | 33.24 | 24.06 |
| | MoD-D | 40.0% | 33.00 | 51.38 | 65.56 | 54.01 | 50.28 | 38.09 | 30.20 | 46.07 | 40.42 | 52.76 | 46.59 |
| | D-LLM | 40.0% | 31.80 | 51.78 | 58.54 | 48.30 | 50.00 | 44.82 | 26.88 | 44.59 | 52.78 | 231.66 | 142.22 |
| | SkipGPT-Joint | 40.3% | 32.94 | 51.23 | 60.41 | 51.24 | 50.21 | 39.75 | 30.76 | 45.22 | 13.28 | 45.63 | 29.46 |
| | SkipGPT-RT-L | 40.3% | **40.80** | **74.98** | **79.16** | **80.33** | **80.00** | **82.74** | **54.01** | **70.29** | **7.70** | **13.10** | **10.40** |

*Table 2.* Comparison of SkipGPT-RT-L against LoRA-finetuned static pruning and dynamic pruning baselines. LoRA results for shortened-PPL, Shortened-Taylor, and SliceGPT on LLaMA2-13B are omitted as these methods fail to converge during recovery training.

the router struggles to refine its decisions, further degrading both its effectiveness and overall model performance. Ultimately, this prevents the joint training paradigm from achieving strong results. In contrast, our two-stage training paradigm effectively mitigates these challenges, enabling the pruned model to achieve superior performance compared to existing dynamic pruning methods.

## 6. Routing Behavior Analysis

Beyond efficiency, we explore if router tuning in SkipGPT-RT reveals deeper insights into the original LLMs. By analyzing the router's behavior, we aim to uncover meaningful patterns in how different modules contribute to inference.

**Comparison of Redundancy between Attention and MLP Modules** Recent studies suggest that LLMs are more resilient to the removal of self-attention layers than feed-forward layers, indicating that the attention modules exhibit higher redundancy compared to MLP modules (Siddiqui et al., 2024; He et al., 2024; He et al., 2025). To explore this further, we analyze five pruned models of LLaMA-2-13B, generated through router tuning under different target sparsity levels $\mathcal{T}$. For each pruned model, we measure the average sparsity of the attention and MLP modules using 50 randomly selected sentences from the WT2 dataset (Merity et al., 2016). Here, sparsity is defined as the ratio of pruned modules to the total number of modules in either the attention or MLP. As shown in Figure 5, SkipGPT-

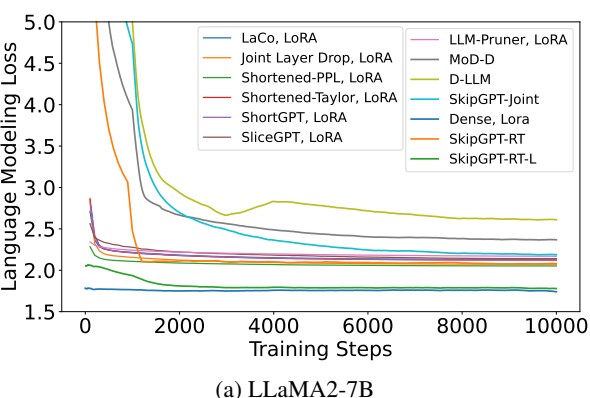
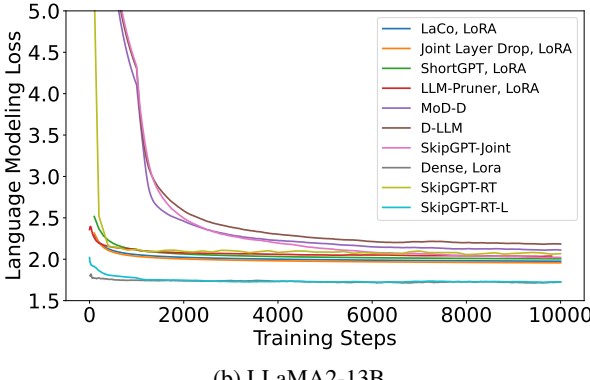

(a) LLaMA2-7B
(b) LLaMA2-13B

*Figure 4.* Training loss curves of LoRA-finetuned static and dynamic pruning baselines, and the two-stage training of SkipGPT.

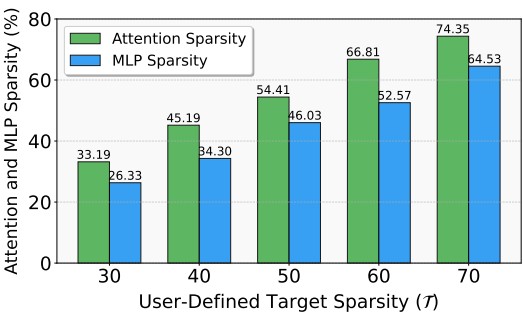

*Figure 5.* Average sparsity of the attention and MLP modules in five pruned models of LLaMA-2-13B, generated through router tuning under different target sparsity levels $\mathcal{T}$.

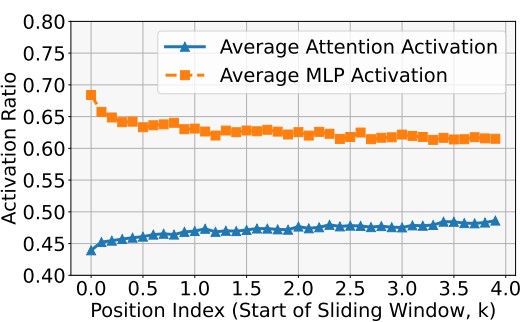

*Figure 6.* Redundancy shifts in attention and MLP modules of a 45% sparsity LLaMA2-13B model obtained via router tuning as context length grows.

RT consistently prunes more attention than MLP modules, across both low and high pruning rates. This observation aligns with previous findings and may indicate a structural inefficiency in the current Transformer architecture, which enforces a fixed pairing of one attention and one MLP module per layer. *We hypothesize that future LLMs could achieve greater efficiency by revisiting this design and potentially reducing the proportion of attention modules.*

**Redundancy Shifts in Attention and MLP Modules** Del Corro et al. (2023) argues that as the context grows with the sequence, later tokens become more predictive and therefore require less computation. Meanwhile, He et al. (2025) shows that the MLP module has higher redundancy during the decoding phase compared to the pre-filling phase, while attention redundancy remains nearly the same in both phases. We investigate how redundancy shifts within the attention and MLP modules by analyzing a 45% sparsity model obtained through router tuning on LLaMA2-13B. We randomly select sentences from RedPajama, truncate them to the model's maximum context length, and apply a 100-token sliding window across each sentence. At each step, we record the activation ratios of attention and MLP within the sliding window and average the results over 50 examples. The final results appear in Figure 6. *Contrary to previous findings, we observe that as context grows, later*

*tokens require more attention computation but less MLP computation.* We hypothesize that in the early stages, the model has not yet determined the task and has limited contextual information. As a result, it relies more on complex MLP computations for task identification and less on attention for context extraction. Later, as the task becomes clear and context accumulates, the model shifts to increased attention computation while reducing MLP computation. Developing techniques that dynamically adjust computation presents an exciting direction for future research.

## 7. Conclusion

In this work, we introduce **SkipGPT**, a novel dynamic pruning framework that addresses the inefficiencies of static layer pruning by incorporating **horizontal and vertical dynamics**. SkipGPT dynamically allocates computation per token and decouples attention and MLP pruning, enabling more fine-grained optimization. To stabilize training, we propose a **two-stage training paradigm**, where the router is first tuned alone before fine-tuning the model with LoRA. Experiments show that the pruned model can **fully restore its performance**, even **surpassing** the original model despite a 40% parameter reduction. Furthermore, our router analysis reveals key insights into **module redundancy and token-level compute allocation**, highlighting potential directions for future model efficiency improvements.

## Impact Statement

Our work, SkipGPT, introduces a novel dynamic pruning framework that significantly improves the efficiency of large language models (LLMs) by adapting computation to token complexity. By decoupling MLP and attention pruning and introducing a two-stage training paradigm, our approach enhances computational efficiency while preserving, and even surpassing, original model performance. This reduces the energy demands of LLMs, making them more sustainable and accessible for deployment in resource-constrained environments, while also providing new insights into model redundancy and architectural optimization.

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

# A. Related Work

**Static Pruning.** Static pruning refers to a kind of approach where *the computation of the pruned LLMs remains invariant to the input instances.* SparseGPT (Frantar & Alistarh, 2023) simplifies the pruning problem by turning it into large-scale sparse regression tasks, which are efficiently solved using a novel approximate solver. FLAP (An et al., 2024) and LLM-Pruner (Ma et al., 2023) reduce network width by eliminating coupled structures while keeping the number of layers unchanged. Sheared-LLaMA (Xia et al., 2023) learns pruning masks at various granularities, from global ones like layers and hidden dimensions to local ones like attention heads and intermediate dimensions. SliceGPT (Ashkboos et al., 2024) reduces the network's embedding dimension by replacing each weight matrix with a smaller dense matrix. Recent works (Song et al., 2024; Gromov et al., 2024; Kim et al., 2024; Chen et al., 2024a; Yang et al., 2024; Siddiqui et al., 2024) demonstrate that it is possible to selectively drop blocks from a range of pretrained language models, sparking community interest in depth pruning. In this context, Zhang et al. (2024) and He et al. (2024) investigate the decoder layers, treating the self-attention layers and FFN layers as independent components to be pruned separately, and observe a preference for pruning the self-attention layers. Despite significant advancements in static pruning, substantial recovery fine-tuning is often necessary to preserve performance post-pruning, making the process both costly and challenging to scale.

**Dynamic Pruning.** Dynamic pruning refers to another kind of approach where *the pruning of unimportant layers is dynamically determined based on the specific input instance.* A common technique in this context is *Early Exit* (Schuster et al., 2022; Varshney et al., 2023; Del Corro et al., 2023; Yom Din et al., 2024; Chen et al., 2024c; Fan et al., 2024). This approach dynamically evaluates whether to continue processing subsequent transformer blocks. Notably, transformer blocks that produce predictions matching the final token output of LLMs are typically located towards the end of the model. As a result, extensive training is often required to adapt LLMs for the effective use of *early exit* mechanisms. Therefore, *early exit* strategies have been rarely explored in larger SOTA LLMs. Another prominent technique is *Skip Layer*, which dynamically skips the execution of intermediate layers (or modules) for a given input token. This is achieved through mechanisms such as a gating function (Wang et al., 2018; Raposo et al., 2024) or a binary router (Zeng et al., 2023; Jiang et al., 2024). For instance, Mixture-of-Depths (MoD) (Raposo et al., 2024) determines which tokens to process using a top-$k$ routing mechanism. Essentially, SkipGPT aligns with the *Skip Layer* paradigm, where the execution of layers for each input token is dynamically determined. To the best of our knowledge, *ours is the first work to simultaneously address both horizontal and vertical dynamics.*

# B. Training Algorithm of SkipGPT

The detailed two-stage training paradigm is outlined in Algorithm 1.

# C. Detailed Descriptions of All Baseline Methods

**ShortGPT** is a structured pruning method that removes redundant layers in LLMs based on a novel importance metric called Block Influence (BI). By analyzing hidden state transformations, ShortGPT assigns BI scores to measure each layer's contribution to model performance. Layers with lower BI scores are identified as redundant and pruned in ascending order. This simple yet effective approach significantly reduces model size and inference cost while maintaining competitive performance. Unlike complex pruning techniques, ShortGPT demonstrates that LLMs contain substantial layer-wise redundancy, enabling efficient compression without additional fine-tuning.

**Shortened LLaMA** is a depth pruning method that reduces the computational cost of LLMs by removing entire Transformer layers while keeping the remaining architecture intact. It determines layer importance using Perplexity (PPL) and Taylor Expansion, pruning less significant layers in a one-shot manner. Unlike width pruning, which reduces weight matrices but struggles to accelerate inference under memory constraints, Shortened LLaMA achieves significant speedups, particularly in resource-limited settings. To recover pruned models, it employs LoRA-based fine-tuning for moderate pruning and Continued Pretraining (CPT) for aggressive pruning. This method provides an efficient way to compress LLMs while maintaining strong performance.

**LaCo (Layer Collapse)** is a structured pruning method that progressively merges deeper layers into earlier ones, reducing model depth while preserving output representations. Instead of removing layers outright, LaCo employs Reserving-Differences-while-Seeking-Common (RDSC) Layer Merge, which integrates parameter differences from consecutive layers

---

**Algorithm 1** Training Process of SkipGPT

---

**Require:** Pretrained model $M$, dataset $D$, target sparsity $\mathcal{T}$, router parameters $\theta$, learning rates $\eta_1$ (router) and $\eta_2$ (LoRA), maximum steps $S_1$ and $S_2$

1: Initialize router parameters $\theta$ for each module
2: Initialize LoRA parameters *(Optional)*
3: **Stage 1: Router Tuning**
4: **for** step $s = 1$ to $S_1$ **do**
5:    $X \sim \text{sample}(D)$ (Sample batch)
6:    **for** token $t$ in $X$ **do**
7:       **for** module $l = 1$ to $L$ **do**
8:          $r_l^t \leftarrow \mathbf{W}_\theta^T x_l^t$ *with $\frac{\partial \mathcal{L}_{all}}{\partial \mathbf{W}_\theta}$ being active*
9:          $g_l^t \sim \text{Gumbel-Softmax}(r_l^t)$
10:         $x_{l+1}^t \leftarrow g_l^t[1] \cdot (f_l(x_l^t) + x_l^t) + g_l^t[0] \cdot x_l^t$
11:       **end for**
12:    **end for**
13:    $r \leftarrow \frac{\sum_{t,l} g_l^t[0]}{S \times L}$
14:    $\mathcal{L}_{\text{all}} \leftarrow \mathcal{L}_{\text{lm}} + \alpha|\mathcal{T} - r|$
15:    Update $\theta \leftarrow \theta - \eta_1 \nabla_\theta \mathcal{L}_{\text{all}}$
16: **end for**
17: **Stage 2: LoRA Fine-Tuning** *(Optional)*
18: **for** step $s = 1$ to $S_2$ **do**
19:    $X \sim \text{sample}(D)$
20:    **for** token $t$ in $X$ **do**
21:       **for** module $l = 1$ to $L$ **do**
22:          $r_l^t \leftarrow \mathbf{W}_\theta^T x_l^t$ *with $\frac{\partial \mathcal{L}_{lm}}{\partial \mathbf{W}_\theta} = 0$ (routers frozen)*
23:          $a_l^t \leftarrow \text{argmax}(r_l^t)$
24:          **if** $a_l^t = 1$ **then**
25:             $x_{l+1}^t \leftarrow f_l(x_l^t) + x_l^t$
26:          **else**
27:             $x_{l+1}^t \leftarrow x_l^t$
28:          **end if**
29:       **end for**
30:    **end for**
31:    Compute $\mathcal{L}_{\text{lm}}$
32:    Update LoRA parameters using $\eta_2$ and $\nabla_{\text{LoRA}} \mathcal{L}_{\text{lm}}$
33: **end for**
**Return:** Pruned model $M'$

---

into a prior layer, maintaining structural integrity. To minimize performance degradation, it uses few-shot calibration samples to ensure representation similarity. This approach enables 30-50% layer reduction without retraining, significantly lowering computational costs while retaining strong performance. Additionally, post-training on pruned models further restores accuracy, making LaCo a highly effective structured pruning technique for large language models.

**Joint Layer Drop** is a structured pruning method that enhances the efficiency of Transformer-based LLMs by jointly pruning Attention and MLP layers based on a similarity-based metric. This method first removes redundant Attention layers, as they exhibit significant redundancy while maintaining performance. Once the least important Attention layers are pruned, it selectively removes MLP layers to further compress the model. By dynamically balancing the pruning of both components, Joint Layer Drop achieves higher compression ratios with minimal accuracy degradation, making it a highly effective approach for structured model reduction.

**LLM-Pruner** is a structured pruning method designed for task-agnostic compression of LLMs, aiming to reduce model size while preserving their general-purpose capabilities. It employs dependency-based structural pruning, where

interdependent components are grouped and pruned together to minimize disruption. Importance estimation is performed using gradient-based and Hessian-based metrics, ensuring the least impactful components are removed. Unlike traditional compression methods that require extensive retraining, LLM-Pruner enables efficient post-training recovery using LoRA-based fine-tuning, requiring only 50K public samples. This approach significantly reduces computational overhead while maintaining strong zero-shot performance across multiple tasks.

**SliceGPT**   is a post-training sparsification method that reduces the computational and memory demands of LLMs by removing entire rows and columns of weight matrices, effectively shrinking the embedding dimension while maintaining dense matrix operations for efficient execution. It leverages computational invariance, applying orthogonal matrix transformations to ensure minimal performance degradation. Using Principal Component Analysis (PCA), SliceGPT projects activation signals onto their principal components, allowing redundant dimensions to be pruned. This method achieves up to 30% compression on LLaMA-2, OPT, and Phi-2 models while retaining over 90% of the original model's accuracy, leading to significant inference speedups and reduced hardware requirements without additional fine-tuning.

**Mixture-of-Depths (MoD)**   is a conditional computation method that dynamically allocates compute across model depth, reducing unnecessary computations in transformer-based LLMs. Unlike standard transformers, which apply the same amount of compute to all tokens at every layer, MoD employs a top-$k$ routing mechanism to selectively process only the most important tokens in each layer, skipping others via residual connections. This allows the model to optimize compute expenditure per token, maintaining performance while significantly reducing FLOPs per forward pass. By ensuring a static computation graph with predictable efficiency gains, MoD achieves up to 50% faster inference while matching or surpassing the performance of equivalent full-compute transformers.

**D-LLM**   is a dynamic inference framework for LLMs that adaptively allocates computational resources at the token level, optimizing efficiency without compromising performance. It introduces a dynamic decision module before each transformer layer, determining whether a token should execute the layer or be skipped, thereby reducing unnecessary computation for less critical tokens and simpler tasks. To maintain compatibility with KV-cache mechanisms, D-LLM employs a KV-cache eviction policy, excluding skipped layers from subsequent attention calculations, which not only reduces storage overhead but also ensures smooth deployment in real-world applications. Experimentally, D-LLM achieves up to 50% reduction in computational cost across various NLP tasks while maintaining strong accuracy, making it a highly effective solution for resource-constrained environments.

## D. Exploring Pruning Scaling Laws with SkipGPT-RT

Due to its significantly superior performance over the baseline, we believe that SkipGPT-RT is capable of identifying optimal or near-optimal routing solutions for a given sparsity ratio. This capability establishes SkipGPT not only as an effective pruning method but also as a reliable probing tool for exploring the pruning scaling laws in existing LLMs. To this end, in this section, we employ SkipGPT-RT as a "probe" to analyze both LLaMA2-7B and LLaMA2-13B, using the SOTA static pruning method, Joint Layer Drop, for comparison. Figure 7 highlights several key findings:

- **Static pruning methods lack scalability:** While static pruning methods are simple and effective in some scenarios, their ability to preserve model performance diminishes significantly under high sparsity levels. For instance, in the case of LLaMA2-13B, SkipGPT-RT demonstrates significantly lower PPL even at 70% sparsity, outperforming Joint Layer Drop, which struggles to maintain comparable performance even at a much lower sparsity of 50%.

- **The surprising redundancy in LLMs:** Our analysis reveals that LLMs exhibit far greater redundancy than expected. For instance, LLaMA2-13B only begins to show a noticeable increase in PPL at an **80**% sparsity level. This phenomenon may stem from two key factors: (1) the inefficiency of the current transformer architecture, which applies uniform computation to all tokens regardless of their importance, and (2) the concentration of critical information within a small subset of modules during pretraining. We hope this finding offers valuable insights for designing future architectures and pretraining strategies.

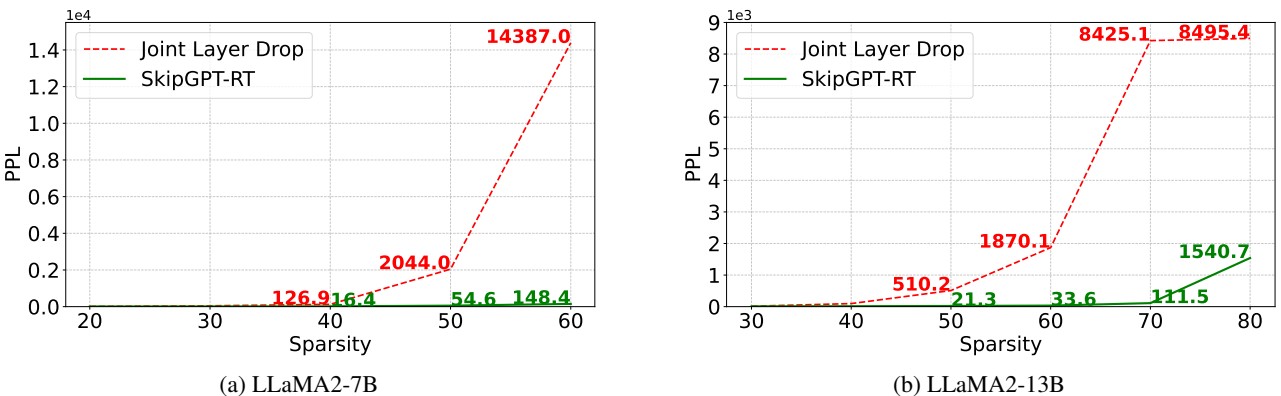

(a) LLaMA2-7B

(b) LLaMA2-13B

*Figure 7.* Perplexity (PPL) of Joint Layer Drop and SkipGPT-RT under different sparsity levels.

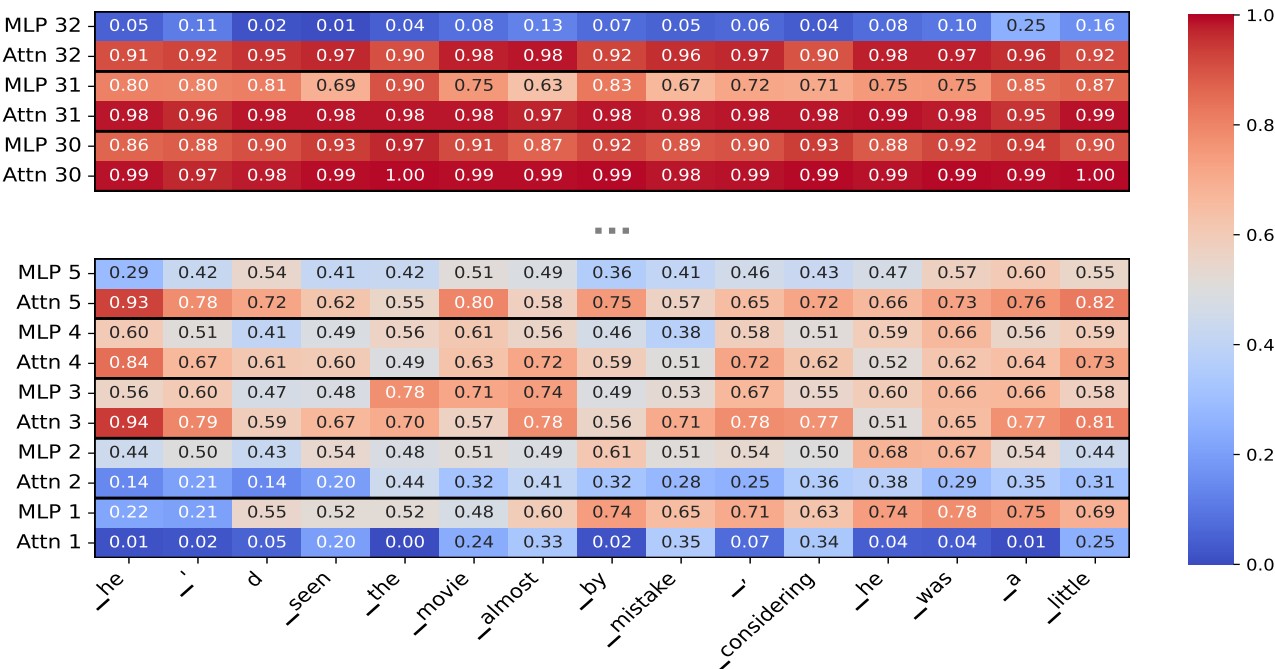

*Figure 8.* Token-Wise Cosine Similarities Across Modules in LLaMA-2-7B.

# E. Additional Case Studies on Motivation

In this section, we present additional case studies that illustrate Token-Wise Cosine Similarities Across Modules in LLaMA2-7B, as shown in Figures 8 and 9. For LLaMA2-13B, see Figures 10 and 11.

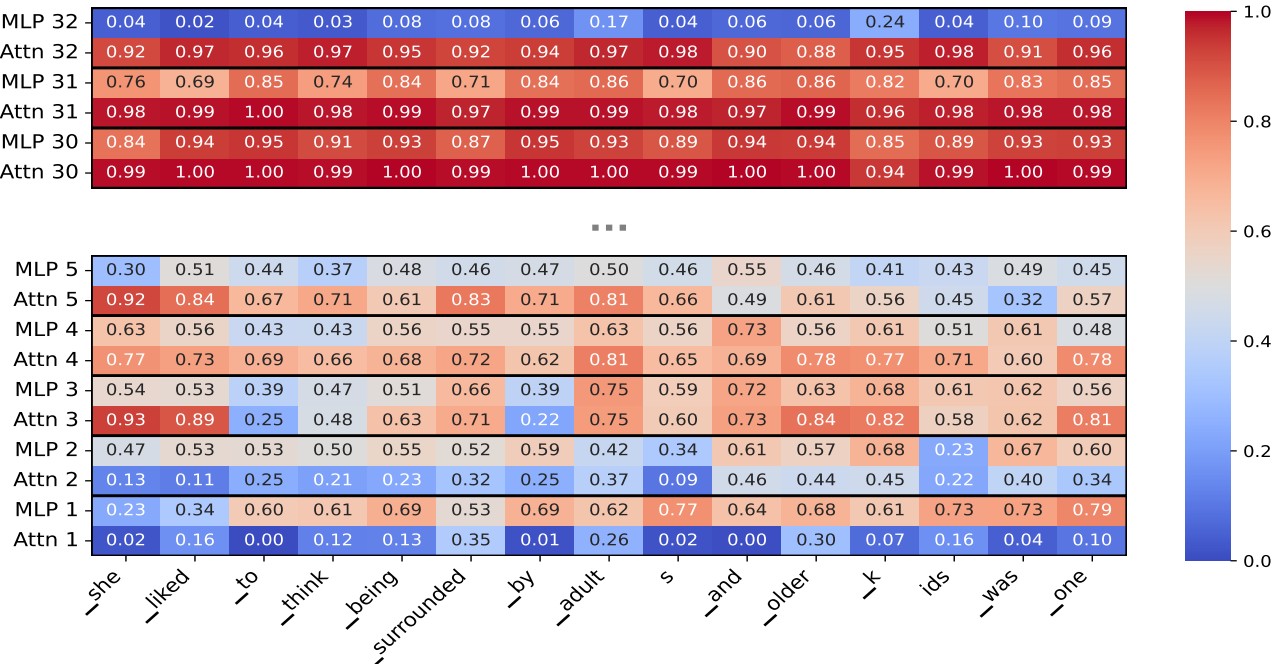

Figure 9. Token-Wise Cosine Similarities Across Modules in LLaMA-2-7B.

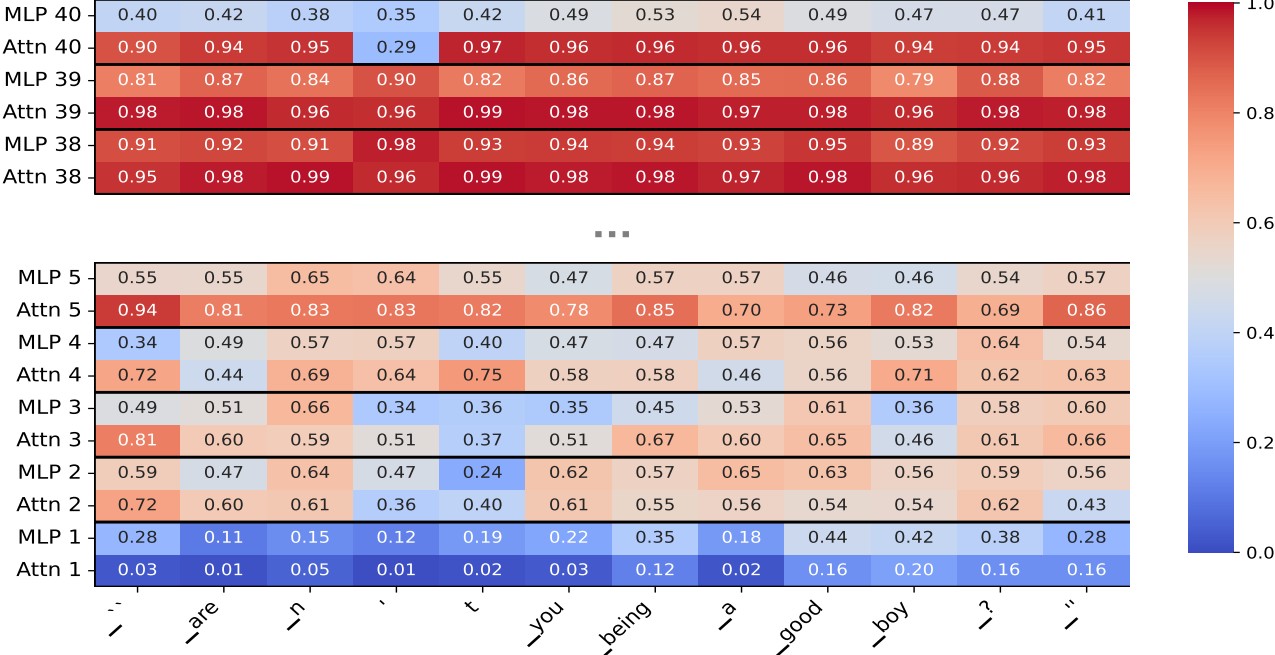

Figure 10. Token-Wise Cosine Similarities Across Modules in LLaMA-2-13B.

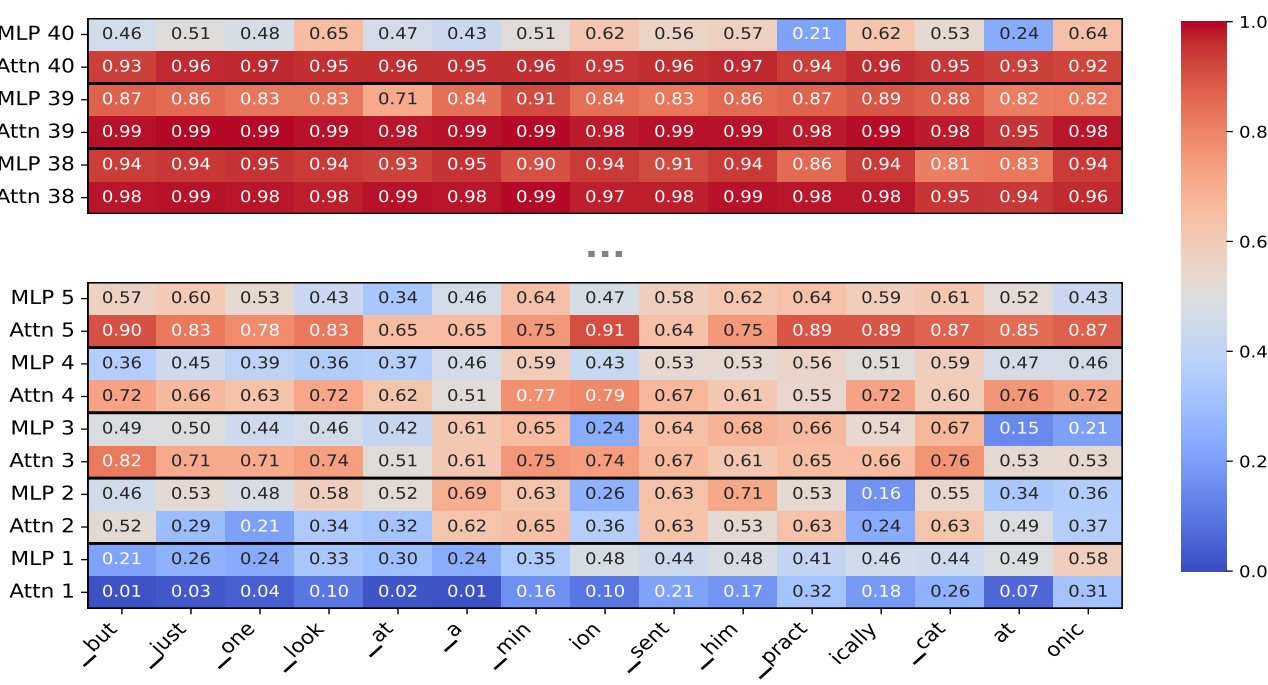

*Figure 11.* Token-Wise Cosine Similarities Across Modules in LLaMA-2-13B.

