# OpenReview forum: "SkipGPT: Each Token is One of a Kind"
_ICML.cc/2025/Conference — ICML 2025 poster_

### Official Review · Reviewer_kVh7 · 2025-03-12

**Overall Recommendation:** 4

**Summary:**

The paper focuses on speeding up inference of large language models (LLMs) by skipping different layers, at the granularity of attention or feed forward network (FFN) layers, for different tokens. It does this by introducing a router (a simple MLP layer with Gumble-Softmax) for each attention and FFN layer. It first trains the router while freezing the model, then introduces LoRA adapters and trains both LoRA and router parameters. The results show that when skipping 25% of parameters, the proposed solution is better than many existing pruning or layer skipping approaches on various tasks and model sizes (Llama2 7B and 13B).

## Update after Rebuttal
I have read all the reviews and rebuttals and I would like to keep my score. With a very limited training budget, the paper exceeds SOTA in layer skipping and the insights and proposal in the paper are important for the academic community.

2 questions just popped to me while doing the final revision are:
1. How KV cache is handled for skipped layers? When token i skips a certain layer but token i+1 runs the same layer, does the query of token i+1 attend to the key and value of token i for that layer?
2. For prefill (that I assume was used to measure perplexity as well as multiple choice tasks like HellaSwag, etc.), were you skipping different layers for different tokens as well?

**Claims And Evidence:**

- Authors made detailed comparison with large number of static and dynamic pruning methods on 2 model sizes and show they perform better than most of them on most tasks
- Authors provided multiple ablation studies to support their claims

**Essential References Not Discussed:**

- The paper made sufficient references and discussed a relatively large number of papers

**Experimental Designs Or Analyses:**

- Paper provided detailed ablation studies to prove the importance of each component of the solution: effectiveness of transformer routing vs attention/ffn routing, effectiveness of two-stage training, routing analysis
- From my previous experience, perplexity is a better evaluation metric of a language model's performance compared to accuracies of so-called commonsense reasoning tasks that are multiple-choice questions. Hence, I would pay more attention to perplexity
- Line 312 (Second Column): "For long contexts, the FLOPs of attention surpass those of the MLP. As a result, SkipGPT-RT achieves lower computational overhead than other baselines." Where in the Tables do we see results of long-context evaluation?
- Please also specify the context length when perplexity was evaluated

**Methods And Evaluation Criteria:**

- Since this paper was submitted in 2025, I believe it should also consist of experiments on Llama3 that was open sourced in April 2024.
I know there is a risk of an accuracy drop because Llama3 is harder to compress but that's fine.
- I also recommend to add results for experiments on generation tasks such as GSM8K, Natural Questions, HumanEval, MBPP

**Other Comments Or Suggestions:**

- Figure 1: "Static Structured Pruning" sub-figure doesn't seem static. Different tokens skip different sub-layers in both "SkipGPT" and "Static Structured Pruning" sub-figures, so they both seem dynamic. In fact, the 2 sub-figures are almost identical to each other.
- Line 190: "Gumbel distribution Gumbel(0, 1)". Is the additional "Gumbel" here a typo?
- Equation 4: I suggest to write down the explicit definitions of y_soft and y_hard
- Line 210 (Right Column): I recommend adding subscript l to W_{theta}
- Line 226: please write the definition of g_l as a function of r_l
- Can you measure end-to-end latency with and without the router?
- Line 246 (Right Column): Please briefly specify in the main body of the paper basic configuration of LoRA finetuning: the rank used, and which weight matrices (Wk, Wv, Wq, Wo, Wup, Wdown?) it was applied on
- For Static Pruning Baselines, you may consider comparing with the pretrained checkpoints of LayerSkip Llama2 7B and Llama2 13B checkpoints ( https://huggingface.co/collections/facebook/layerskip-666b25c50c8ae90e1965727a ) by removing the last 25% of the layers
- Figure 4: Too many curves with similar colors. To make it easier, I suggest to order the plots in the legend in the same order as the Language Modeling Loss in the last training step

**Other Strengths And Weaknesses:**

- Strengths:
  - Novelty: "To overcome this limitation, we introduce a novel sparsity concept that defines computation budgets across the entire forward pass, rather than being confined to layer-wise or token-wise constraints"
  - Approach:
     - Very lightweight router
     - Easy and fast: requires a single A800 GPU and 4 hours of finetuning
  - Useful insights presented, such as :
    - "(1) Attention modules exhibit greater redundancy than MLP modules. (2) As context length increases, later tokens demand more attention computation but less MLP processing."
    - Instability introduced by joint training of untrained routers with trained parameters
  - Writing style:
    - Authors build the motivation of their approach in an intriguing and clear way.
    - Each component of the solution is explained in a clear and detailed manner to build up the full picture

**Questions For Authors:**

- Table 1: Does the ratio of parameters used put into consideration the router and LoRA parameter count?

**Relation To Broader Scientific Literature:**

- The paper made a concise summary of related work and compared with a large number of static layer pruning, dynamic layer pruning, as well as embedding dimension pruning

**Theoretical Claims:**

N/A

---

> ### Author Rebuttal · Authors · 2025-04-01
>
> **Q1**: The reviewer suggests including experiments on LLaMA3, which was released in April 2024.
>
> **Response**:  To address the reviewer’s suggestion, we conducted additional experiments on LLaMA3-8B, evaluating SkipGPT under both 25% and 40% sparsity settings. Due to space constraints, we compare SkipGPT with two of the most competitive baselines: ShortGPT and Shortened LLaMA (PPL). The table below presents the results under 25% sparsity, comparing SkipGPT-RT with the baselines across several downstream tasks and language modeling benchmarks:
>
> |Model|OpenBookQA|Winogrande|PIQA|HellaSwag|BoolQ|ARC-Easy|ARC-Challenge|Avg. Acc|WikiText|PTB|Avg. PPL|
> |-|-|-|-|-|-|-|-|-|-|-|-|
> |Dense|44.8|77.51|80.03|81.95|82.14|84.85|57.59|72.69|6.24|10.58|8.41|
> |ShortGPT|29.2|52.25|60.44|30.80|37.64|36.70|30.80|39.69|2796.24|2799.46|2797.85|
> |SkipGPT|44.2|75.69|78.07|76.87|74.06|82.44|53.41|69.25|16.47|26.91|21.69|
> |Shortened LLaMA (PPL)|33.6|58.64|71.54|55.94|39.69|59.81|31.48|50.10|15.00|23.86|19.43|
>
> As shown, static pruning methods like ShortGPT suffer significant degradation—e.g., ShortGPT shows extremely poor language modeling (PPL > 2700). In contrast, SkipGPT-RT maintains performance close to the dense model, highlighting the importance of dynamic pruning for large, well-trained models like LLaMA3-8B, where full layer removal is no longer viable.
> We observe the same trend under 40% sparsity, where SkipGPT continues to outperform static baselines by a large margin:
>
> |Model|OpenBookQA|Winogrande|PIQA|HellaSwag|BoolQ|ARC-Easy|ARC-Challenge|Avg. Acc|WikiText|PTB|Avg. PPL|
> |-|-|-|-|-|-|-|-|-|-|-|-|
> |Dense|44.8|77.51|80.03|81.95|82.14|84.85|57.59|72.69|6.24|10.58|8.41|
> |ShortGPT|29.1|53.51|60.28|35.49|56.02|34.56|30.34|42.76|79856.66|125507.27|102681.97|
> |SkipGPT|38.0|59.35|73.34|64.36|60.37|77.53|45.65|59.80|71.25|48.05|59.65|
> |Shortened LLaMA (PPL)|31.3|57.45|62.35|50.52|36.93|53.76|31.45|46.23|157.01|196.04|176.53|
>
> Here, ShortGPT fails completely as a language model, producing meaningless outputs. In contrast, SkipGPT-RT maintains over 80% of the dense model’s performance, despite having no fine-tuning beyond router tuning.
> For the LoRA fine-tuning phase, Shortened LLaMA (PPL) fails to converge and is thus omitted. We compare SkipGPT-RT-L (router tuning + LoRA) with ShortGPT under 25% sparsity:
>
> |Model|OpenBookQA|Winogrande|PIQA|HellaSwag|BoolQ|ARC-Easy|ARC-Challenge|Avg. Acc|WikiText|PTB|Avg. PPL|
> |-|-|-|-|-|-|-|-|-|-|-|-|
> |Dense LoRA|44.9|77.68|80.22|81.86|82.23|84.92|57.89|72.81|6.13|10.44|8.29|
> |ShortGPT|37.8|74.27|72.63|70.53|71.19|69.21|47.78|63.34|11.13|16.64|13.89|
> |SkipGPT-RT-L|42.6|77.03|79.97|82.13|82.84|84.47|57.08|72.30|7.10|11.70|9.40|
>
> SkipGPT-RT-L effectively recovers nearly all the performance of the dense model after LoRA adaptation.
> This conclusion holds under 40% sparsity as well:
>
> |Model|OpenBookQA|Winogrande|PIQA|HellaSwag|BoolQ|ARC-Easy|ARC-Challenge|Avg. Acc|WikiText|PTB|Avg. PPL|
> |-|-|-|-|-|-|-|-|-|-|-|-|
> |Dense LoRA|44.9|77.68|80.22|81.86|82.23|84.92|57.89|72.81|6.13|10.44|8.29|
> |ShortGPT|31.0|69.13|67.57|67.24|65.84|62.31|37.20|57.18|18.35|30.65|24.50|
> |SkipGPT-RT-L|40.8|74.98|79.16|80.33|80.00|82.74|54.01|70.29|7.70|13.10|10.40|
>
> **Q2**: The reviewer recommends including results on generation tasks to further evaluate the effectiveness of the method.
>
> **Response**: Thank you for the helpful suggestion. To evaluate our method on generation tasks, we conducted zero-shot experiments on GSM8K (flexible match) and MBPP using LLaMA3-8B under a 40% sparsity setting. We compared SkipGPT-RT-L (router tuning + LoRA) with a strong baseline ShortGPT + LoRA, using the same sparsity level. The results are as follows:
>
> |Models|GSM8K (%)|MBPP (%)|
> |-|-|-|
> |LLaMA-3.1-8B|26.23|30.6|
> |SkipGPT-RT-L|15.34|21.5|
> |ShortGPT + LoRA|3.42|2.12|
>
> While there is some drop in performance compared to the dense model, SkipGPT still significantly outperforms the pruning baseline, demonstrating its advantage in generation tasks.
>
> **Q3:** Can you measure end-to-end latency with and without the router?
>
> **Response:** Please refer to our response to Reviewer hvhB Q2. The router adds negligible overhead, as shown in our end-to-end latency and module-level breakdown. We’ll highlight this in the revision.
>
> **Q4:** The reviewer asks for clarification on the LoRA configuration used in the experiments.
>
> **Response:** Thank you for the suggestion. In our experiments, we use a LoRA rank of 16, applied to \( W_q \), \( W_v \), and \( W_{\text{gate}} \). We will clearly specify these details in the main text in the revision.
>
> **Q5:** Do Table 1 ratios include router/LoRA parameters?
>
> **Response:** Table 1 includes router parameters. Table 2 includes both router and LoRA. We'll clarify in the revision.

---

> > ### Comment · Reviewer_kVh7 · 2025-04-05
> >
> > I would like to thank the authors for their detailed response.
> > Can the authors explain the difference between the first 2 tables in their rebuttal?

---

> > > ### Author Response · Authors · 2025-04-05
> > >
> > > Thank you very much for your thoughtful question and for taking the time to carefully review our rebuttal.
> > >
> > > To clarify: the first table in our response presents the performance of **SkipGPT-RT with a 25% parameter reduction**, obtained through router tuning on the dense model, **compared with baseline methods that apply the same level of parameter reduction**. The second table provides a similar comparison under a **40% parameter reduction** setting.
> > >
> > > Both sets of experiments are conducted on the **LLaMA 3.1 8B** model. We would also like to note that a more comprehensive set of results on LLaMA 3.1 8B will be included in the revision to provide a fuller picture.
> > >
> > > We sincerely appreciate your interest and hope this clarifies the distinction between the two tables. Please feel free to let us know if there is anything further we can elaborate on.

---

### Official Review · Reviewer_GDzU · 2025-03-14

**Overall Recommendation:** 2

**Summary:**

The authors propose SkipGPT by addressing the challenges of existing dynamic pruning methods, namely horizontal dynamics, vertical dynamics, and the training paradigm. In other words, they redefine sparsity, separate MLP and self-attention within layers, and train routing and LoRA in a two-stage process.

**Claims And Evidence:**

Yes, but certain parts appear to be somewhat overextended.

- For instance, The phrase "novel sparsity concept" seems exaggerated. It appears to be merely a removal of certain constraints. In the field that deals with token eviction, it is already standard practice to consider the token axis as a whole. Moreover, the papers like pyramidKV [1] even addresses the layer axis in relation to the overall budget.


---
[1] Cai, Zefan, et al. "Pyramidkv: Dynamic kv cache compression based on pyramidal information funneling." arXiv preprint arXiv:2406.02069 (2024).

**Essential References Not Discussed:**

Please refer to "Claims and Evidence"

**Experimental Designs Or Analyses:**

- Please provide the results of recent LLMs such as llama-3, gemma-2, etc.
- I understand that the structured pruning method maintains meaningful performance up to approximately 30%. However, it would be preferable if it could deliver performance at even higher ratios.
- It would be beneficial to include a comparison of the performance for long context like LongBench.
- Please specify the actual execution time of each algorithm (beyond the theoretical ratio).

**Methods And Evaluation Criteria:**

Yes.

**Other Comments Or Suggestions:**

None

**Other Strengths And Weaknesses:**

None

**Questions For Authors:**

None

**Relation To Broader Scientific Literature:**

This is a study related to the lightweight optimization of large language models (LLMs). But for me, proposed three approach (redefined sparsity, decoupling, two-stage training) are not novel, but it is practical.

**Theoretical Claims:**

None.

---

> ### Author Rebuttal · Authors · 2025-04-01
>
> **Q1**: The phrase "novel sparsity concept" seems exaggerated.
>
> **Response**: We appreciate the reviewer’s feedback regarding the “novel sparsity concept.” Our intention was not to overstate novelty in this regard. Rather, we aimed to highlight that SkipGPT enables fully dynamic layer skipping, where both the number of layers each token passes through and the participating tokens in each module can vary dynamically across the sequence.
>
>  That said, we agree that similar ideas of global token- and layer-level budget allocation have been explored in prior works such as PyramidKV. In our case, this sparsity definition is primarily used as a control mechanism during training to specify and regulate the sparsity target, and is not the main technical contribution of our paper. We will revise the wording in the final version to better reflect this and avoid overstating novelty.
>
> **Q2:** Can the method be evaluated on recent LLMs such as LLaMA-3 or Gemma-2?
>
> **Response:** Thank you for the suggestion. We refer you to our response to Reviewer kVh7 Q1, where we present detailed results and analysis on LLaMA3-8B.
>
> **Q3**: Does the method maintain strong performance under higher pruning ratios beyond 30%?
>
> **Response:** We appreciate the reviewer’s attention to the method’s robustness under higher pruning ratios. While we have already demonstrated in Q2 that SkipGPT remains effective at pruning ratios beyond 30% using LLaMA3-8B, we further conducted a comprehensive set of experiments on LLaMA2-7B, the model originally used in our paper, under a 40% sparsity setting.
>
> The results after router tuning are shown below:
>
> |Model|OpenBookQA|Winogrande|PIQA|HellaSwag|BoolQ|ARC-Easy|ARC-Challenge|Avg.Acc|WikiText|PTB|Avg.PPL|
> |------------------------|------------|------------|-------|-----------|--------|-----------|----------------|-----------|----------|---------|-----------|
> |Dense|44.2|74.19|78.07|78.93|71.62|81.36|52.47|68.69|5.47|20.83|13.15|
> |ShortGPT|34.4|62.83|59.58|45.43|62.17|44.82|30.80|48.58|79.47|156.67|118.07|
> |JointLayerDrop|30.2|57.85|60.61|41.32|62.17|36.91|30.72|45.68|126.88|302.04|214.46|
> |LLM-Pruner|32.2|53.35|65.67|42.41|59.97|32.24|26.37|44.60|46.34|191.31|118.83|
> |SkipGPT-RT|34.2|55.72|63.87|51.09|56.30|56.94|31.83|50.00|16.69|91.00|53.85|
>
>
> We observe that our method outperforms all pruning baselines on both downstream accuracy and language modeling perplexity under 40% sparsity, demonstrating its superior adaptability and robustness. To further validate effectiveness, we also apply LoRA fine-tuning under the same 40% sparsity setting. The results are summarized below:
>
> |Model|OpenBookQA|Winogrande|PIQA|HellaSwag|BoolQ|ARC-Easy|ARC-Challenge|Avg.Acc|WikiText|PTB|Avg.PPL|
> |------------------|------------|------------|-------|-----------|--------|-----------|----------------|-----------|----------|---------|-----------|
> |DenseLoRA|44.8|74.27|78.02|78.96|79.02|81.73|53.07|69.98|5.48|20.58|13.03|
> |ShortGPT,LoRA|34.0|65.90|64.91|57.30|63.46|55.60|33.02|53.46|14.78|49.71|32.25|
> |JointLayerDrop,LoRA|35.4|63.22|69.42|62.06|69.91|62.12|36.18|56.90|11.08|42.19|25.64|
> |LLM-Pruner,LoRA|32.2|53.51|65.72|42.42|60.00|32.24|26.37|44.64|46.33|191.31|118.82|
> |SkipGPT-RT-L|43.0|72.93|77.09|76.63|76.88|81.48|52.39|68.63|6.00|31.05|18.53|
>
> As shown, after LoRA fine-tuning, our method nearly fully recovers the performance of the dense model.
>
> **Q4:** How does the method perform on long-context benchmarks such as LongBench?
>
> **Response:** Thank you for the question. To evaluate our method on long-context understanding, we conducted experiments on LongBench. Specifically, we applied router tuning followed by supervised fine-tuning (SFT) on the LLaMA 3.1 8B base model using the Alpaca dataset. We compare our model (SkipGPT-RT-SFT, with 40% sparsity) against the official LLaMA-3.1-8B-Instruct, as well as a strong pruning baseline ShortGPT, which uses the same SFT configuration as our method. The following table summarizes the results (without CoT prompting):
>
> |Model|Overall(%)|Easy(%)|Hard(%)|Short(%)|Medium(%)|Long(%)|
> |----------------------|-------------|----------|----------|------------|--------------|-----------|
> |LLaMA-3.1-8B-Instruct|29.8|30.7|29.6|35.0|27.9|25.9|
> |SkipGPT-RT-SFT|28.6|29.3|28.3|32.7|26.5|26.3|
> |ShortGPT+SFT|25.5|25.7|26.1|24.3|26.3|25.3|
>
> Although LongBench remains a challenging benchmark, our method significantly outperforms the pruning baseline (ShortGPT) across all difficulty levels and context lengths. While SkipGPT-RT-SFT does not fully match the dense model’s performance, it shows strong capability in long-context scenarios—even under 40% sparsity—highlighting the effectiveness of our compressed approach.
>
> **Q5**: Can you provide actual execution time comparisons rather than just theoretical speedup ratios?
>
> **Response:** Thank you for your question. Please refer to our response to Reviewer hvhB Q2.

---

### Official Review · Reviewer_hvhB · 2025-03-17

**Overall Recommendation:** 3

**Summary:**

The paper proposes a method to prune LLMs in horizontal (per token processing) and vertical (layer-wise) dimensions. It also provides a two-stage pruning pipeline in which first it trains a router given a fixed pre-trained LLM and then fine-tunes the model with LoRA adapters to recover the performance. Experiments with various benchmarks demonstrate the effectiveness of the proposed method.

**Claims And Evidence:**

Yes, the claims about the need for vertical, horizontal, and two-staged pruning pipeline are properly supported by the experiments.

**Essential References Not Discussed:**

I don't know any essential reference not discussed.

**Experimental Designs Or Analyses:**

Yes, the paper experiments with two variants of LLaMA models on well-known benchmarks.

**Methods And Evaluation Criteria:**

* I think definition of sparsity in the paper is problematic. In Eq. (7), the number of active blocks determines sparsity. However, there are more practical and principled metrics of sparsity like number of active parameters in MoE models or MACs/FLOPs in the model pruning literature that can better reflect inference efficiency.

* Following the previous point, if the definition of in Tab. 1 be the same as Eq. (7), it makes it difficult to determine practical usefulness of the proposed method. Although the idea of fine-grained resource allocation to tokens is intuitive, it is usually hard to gain proportional inference latency speed up due to the required slicing operations in practice. Also, the inference latency for different blocks like attention vs MLPs depends on factors like the number of tokens (like the fact that attention is quadratic w.r.t the sequence length) and the software package (for instance Flash Attention is highly efficient on H100 GPUs) for the implementation. Therefore, I believe that the paper needs to report inference latency values for the SkipGPT vs the dense baseline to better demonstrate the practical significance of the proposed method.

**Other Comments Or Suggestions:**

I recommend that the paper provide inference latency numbers for the pruned model vs the baseline dense model, and if possible with baseline methods. As I mentioned above, the reduction in per token computation may not necessarily lead to proportional inference latency reduction. I will raise my score if the authors can provide compelling evidence in this regard.

**Other Strengths And Weaknesses:**

Please check the sections above.

**Questions For Authors:**

Please check the sections above.

**Relation To Broader Scientific Literature:**

The paper makes the pruning more fine-grained in terms of depth pruning and token processing. On the depth pruning side, it prunes the attention layers and MLP layers in the transformer blocks separately. On the token processing side, it selects whether each token be processed by a layer or not.

**Theoretical Claims:**

The paper has no theoretical claims.

---

> ### Author Rebuttal · Authors · 2025-04-01
>
> **Q1:** Is the definition of sparsity in the paper appropriate and reflective of practical inference efficiency?
>
> **Response:** Our sparsity metric, based on skipped modules, is a simple and consistent proxy for dynamic pruning, though it doesn't directly reflect FLOPs. While actual efficiency depends on sequence length and module type, MLP and attention FLOPs are roughly comparable on average (e.g., in LLaMA2-7B), making our proxy a reasonable simplification. For fair comparisons, we match parameter sparsity with the pruning baseline. Since attention has fewer parameters and is skipped more often, this leads to lower actual FLOPs—especially for long sequences. We plan to include FLOP and latency metrics in future versions.
>
> **Q2:** Can the proposed method demonstrate real-world inference speedup, and are latency measurements compared to dense baselines reported?
>
> **Response:** Thank you for highlighting this important point. In practice, LLM inference consists of two phases: prefilling (processing the initial prompt) and decoding (generating tokens step by step).
>
> **(1) Prefilling phase**
>
> To quantify the latency overhead introduced by our method, we conduct detailed timing analysis on an A800 GPU using a SkipGPT model (LLaMA2-7B) with 25% sparsity. For an 80-token input, the additional operations per layer introduced by SkipGPT include attention/MLP routing, argmax, and slicing. The average per-layer latency (in seconds) is as follows:
> | Operation                          | Avg. Time per Layer (s) |
> |-----------------------------------|--------------------------|
> | Attention Router                  | 0.000314                 |
> | Argmax                            | 0.000154                 |
> | MLP Router                        | 0.000202                 |
> | Slicing                           | 0.000140                 |
> | Grouped Computation (post-slice) | 0.057000                 |
> | **Total (SkipGPT)**              | **0.057964**             |
> | **Total (Dense)**                | **0.064000**             |
>
> This yields an average ~10% reduction in per-layer latency during prefilling. We also report end-to-end prefilling latency across sequence lengths for both dense and 25% sparse models, showing the sparse model’s runtime as a fraction of the dense model’s:
> | Input Tokens | Dense (s) | SkipGPT (25%) (s) | Ratio (SkipGPT / Dense) |
> |--------------|-----------|-------------------|--------------------------|
> | 80           | 2.048     | 1.854             | 0.905                    |
> | 800          | 3.964     | 3.369             | 0.850                    |
> | 2000         | 22.618    | 20.862            | 0.922                    |
>
> **(2) Decoding phase**
>
> Due to token-by-token generation, SkipGPT often achieves greater-than-theoretical speedups. FlashAttention’s per-step overheads—like linear projections, RoPE, KV cache updates, and kernel setup—can’t be amortized, making attention the dominant cost. Since our router frequently skips attention, SkipGPT yields significant latency gains. The table below shows results for a 25% sparsity SkipGPT model (LLaMA2-7B) with an 80-token prompt.
> | Generated Tokens | Dense (s) | SkipGPT (s) | Theoretical Ratio | Actual Ratio |
> |------------------|-----------|-------------|--------------------|--------------|
> | 40   | 5.688| 4.402 | 0.75 | 0.77|
> | 200 | 13.298| 9.920       | 0.75  | 0.75         |
> | 800| 37.089    | 26.383      | 0.75 | 0.71         |
>
> As shown above, the actual speedup closely matches or even exceeds the theoretical upper bound, particularly as the sequence length increases.
>
> **(3) Hardware-aware Optimization (Ongoing FPGA Work)**
>
> We recognize that current GPUs struggle with token-wise dynamic sparsity due to memory bottlenecks and kernel overhead. To address this, we're developing a custom FPGA backend for SkipGPT that separates compute- and memory-bound phases for targeted optimization:
> - Prefilling (Compute-bound): We use a dataflow PE array with sparsity-aware scheduling to boost sparse matrix throughput and reduce idle cycles.
>
>
> - Decoding (Memory-bound): We design a hierarchical memory system with local buffers, lightweight scheduling, and KV cache-aware prefetching, outperforming GPU’s unified memory pipeline.
>
> Using a 25% sparse SkipGPT (LLaMA2-7B), our FPGA prototype achieves the following normalized speedups:
> | Input Tokens | Relative Time |
> |--------------|----------------|
> | 40 | 0.753|
> | 200 | 0.732 |
> | 800| 0.715  |
>
> For a fixed prompt length of 80 tokens, we measure the total decoding time for generating different output lengths:
> | Output Tokens | Relative Time |
> |---------------|----------------|
> | 40 | 0.723 |
> | 200 | 0.719|
> | 800 | 0.712 |
>
> Lastly, we have shown that the sparsity ratio can be further reduced to **40%** without significantly copromising the model performance (See our response to RGDzU Q3 and Reviewer kVh7 Q1). We will add latency measurements and provide results under higher sparsity ratio in the revision.

---

> > ### Comment · Reviewer_hvhB · 2025-04-08
> >
> > I thank the authors for their efforts for the rebuttal. The rebuttal addressed my concerns, and I raise my score.

---

> > > ### Author Response · Authors · 2025-04-09
> > >
> > > We are very grateful to the reviewer for the constructive comments and for taking the time to re-evaluate our work. We truly appreciate your thoughtful engagement and your updated score.

---

### Decision · Program_Chairs · 2025-05-01

**Decision:**

Accept (poster)

**Comment:**

This paper introduces a dynamic pruning framework for LLMs that addresses three key challenges: redefining sparsity through global optimization across token and layer dimensions, decoupling MLP and attention components for selective pruning, and implementing a two-stage training paradigm with router optimization followed by LoRA fine-tuning.

During the rebuttal period, reviewers raised concerns about the sparsity definition, practical efficiency measurements, evaluations on recent models, performance at higher sparsity ratios, and long-context capabilities. The authors responded comprehensively with detailed latency measurements showing reduction in prefill and decoding latency, extensive experiments on Llama 8B demonstrating superior performance over baselines, evaluations on Long-Bench showing strong long-context understanding, and additional tests on generation tasks like GSM8K. Reviewer hvhB updated their score positively.

Key strengths include its novel integration of different pruning strategies, strong empirical results across multiple model sizes and tasks, measurable efficiency gains in real-world scenarios, lightweight training requirements, comprehensive ablations, and valuable insights into component redundancy. While the approach shows limitations at higher sparsity ratios, builds upon some existing techniques, and initially lacked evaluations on newer models, it ultimately represents a good paper worth sharing with the broader community.